# OPTIMISTIC EXPLORATION EVEN WITH A PESSIMISTIC INITIALISATION

**Tabish Rashid, Bei Peng, Wendelin Böhmer, Shimon Whiteson**
University of Oxford
Department of Computer Science
`{tabish.rashid, bei.peng,`
`wendelin.boehmer, shimon.whiteson}@cs.ox.ac.uk`

## ABSTRACT

Optimistic initialisation is an effective strategy for efficient exploration in reinforcement learning (RL). In the tabular case, all provably efficient model-free algorithms rely on it. However, model-free deep RL algorithms do not use optimistic initialisation despite taking inspiration from these provably efficient tabular algorithms. In particular, in scenarios with only positive rewards, $Q$-values are initialised at their lowest possible values due to commonly used network initialisation schemes, a *pessimistic initialisation*. Merely initialising the network to output optimistic $Q$-values is not enough, since we cannot ensure that they remain optimistic for novel state-action pairs, which is crucial for exploration. We propose a simple count-based augmentation to pessimistically initialised $Q$-values that separates the source of optimism from the neural network. We show that this scheme is provably efficient in the tabular setting and extend it to the deep RL setting. Our algorithm, Optimistic Pessimistically Initialised $Q$-Learning (OPIQ), augments the $Q$-value estimates of a DQN-based agent with count-derived bonuses to ensure optimism during both action selection and bootstrapping. We show that OPIQ outperforms non-optimistic DQN variants that utilise a pseudocount-based intrinsic motivation in hard exploration tasks, and that it predicts optimistic estimates for novel state-action pairs.

## 1 INTRODUCTION

In reinforcement learning (RL), exploration is crucial for gathering sufficient data to infer a good control policy. As environment complexity grows, exploration becomes more challenging and simple randomisation strategies become inefficient.

While most provably efficient methods for tabular RL are model-based (Brafman and Tennenholtz, 2002; Strehl and Littman, 2008; Azar et al., 2017), in deep RL, learning models that are useful for planning is notoriously difficult and often more complex (Hafner et al., 2019) than model-free methods. Consequently, model-free approaches have shown the best final performance on large complex tasks (Mnih et al., 2015; 2016; Hessel et al., 2018), especially those requiring hard exploration (Bellemare et al., 2016; Ostrovski et al., 2017). Therefore, in this paper, we focus on how to devise model-free RL algorithms for efficient exploration that scale to large complex state spaces and have strong theoretical underpinnings.

Despite taking inspiration from tabular algorithms, current model-free approaches to exploration in deep RL do not employ *optimistic initialisation*, which is crucial to provably efficient exploration in all model-free tabular algorithms. This is because deep RL algorithms do not pay special attention to the initialisation of the neural networks and instead use common initialisation schemes that yield initial $Q$-values around zero. In the common case of non-negative rewards, this means $Q$-values are initialised to their lowest possible values, i.e., a *pessimistic initialisation*.

While initialising a neural network optimistically would be trivial, e.g., by setting the bias of the final layer of the network, the uncontrolled generalisation in neural networks changes this initialisation quickly. Instead, to benefit exploration, we require the $Q$-values for novel state-action pairs must remain high until they are explored.

An empirically successful approach to exploration in deep RL, especially when reward is sparse, is *intrinsic motivation* (Oudeyer and Kaplan, 2009). A popular variant is based on *pseudocounts* (Bellemare et al., 2016), which derive an intrinsic bonus from approximate visitation counts over states and is inspired by the tabular MBIE-EB algorithm (Strehl and Littman, 2008). However, adding a positive intrinsic bonus to the reward yields optimistic $Q$-values only for state-action pairs that have already been chosen sufficiently often. Incentives to explore unvisited states rely therefore on the generalisation of the neural network. Exactly how the network generalises to those novel state-action pairs is unknown, and thus it is unclear whether those estimates are optimistic when compared to nearby visited state-action pairs.

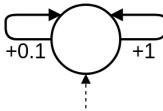

Figure 1

Consider the simple example with a single state and two actions shown in Figure 1. The left action yields $+0.1$ reward and the right action yields $+1$ reward. An agent whose $Q$-value estimates have been zero-initialised must at the first time step select an action randomly. As both actions are underestimated, this will increase the estimate of the chosen action. Greedy agents always pick the action with the largest $Q$-value estimate and will select the same action forever, failing to explore the alternative. Whether the agent learns the optimal policy or not is thus decided purely at random based on the initial $Q$-value estimates. This effect will only be amplified by intrinsic reward.

To ensure optimism in unvisited, novel state-action pairs, we introduce Optimistic Pessimistically Initialised $Q$-Learning (OPIQ). OPIQ does not rely on an optimistic initialisation to ensure efficient exploration, but instead augments the $Q$-value estimates with count-based bonuses in the following manner:

$$Q^+(s,a) := Q(s,a) + \frac{C}{(N(s,a)+1)^M}, \tag{1}$$

where $N(s,a)$ is the number of times a state-action pair has been visited and $M, C > 0$ are hyperparameters. These $Q^+$-values are then used for both action selection and during bootstrapping, unlike the above methods which only utilise $Q$-values during these steps. This allows OPIQ to maintain optimism when selecting actions and bootstrapping, since the $Q^+$-values can be optimistic even when the $Q$-values are not.

In the tabular domain, we base OPIQ on UCB-H (Jin et al., 2018), a simple online $Q$-learning algorithm that uses count-based intrinsic rewards and optimistic initialisation. Instead of optimistically initialising the $Q$-values, we pessimistically initialise them and use $Q^+$-values during action selection and bootstrapping. Pessimistic initialisation is used to enable a worst case analysis where all of our $Q$-value estimates underestimate $Q^*$ and is **not** a requirement for OPIQ. We show that these modifications retain the theoretical guarantees of UCB-H.

Furthermore, our algorithm easily extends to the Deep RL setting. The primary difficulty lies in obtaining appropriate state-action counts in high-dimensional and/or continuous state spaces, which has been tackled by a variety of approaches (Bellemare et al., 2016; Ostrovski et al., 2017; Tang et al., 2017; Machado et al., 2018a) and is orthogonal to our contributions.

We demonstrate clear performance improvements in sparse reward tasks over 1) a baseline DQN that just uses intrinsic motivation derived from the approximate counts, 2) simpler schemes that aim for an optimistic initialisation when using neural networks, and 3) strong exploration baselines. We show the importance of optimism during action selection for ensuring efficient exploration. Visualising the predicted $Q^+$-values shows that they are indeed optimistic for novel state-action pairs.

## 2 BACKGROUND

We consider a Markov Decision Process (MDP) defined as a tuple $(\mathcal{S}, \mathcal{A}, P, R)$, where $\mathcal{S}$ is the state space, $\mathcal{A}$ is the discrete action space, $P(\cdot|s,a)$ is the state-transition distribution, $R(\cdot|s,a)$ is the distribution over rewards and $\gamma \in [0,1)$ is the discount factor. The goal of the agent is then to maximise the expected discounted sum of rewards: $\mathbb{E}[\sum_{t=0}^{\infty} \gamma^t r_t | r_t \sim R(\cdot|s_t, a_t)]$, in the discounted episodic setting. A policy $\pi(\cdot|s)$ is a mapping from states to actions such that it is a valid probability distribution. $Q^\pi(s,a) := \mathbb{E}[\sum_{t=0}^{\infty} \gamma^t r_t | a_t \sim \pi(\cdot|s_t)]$ and $Q^* := \max_\pi Q^\pi$.

Deep $Q$-Network (DQN) (Mnih et al., 2015) uses a nonlinear function approximator (a deep neural network) to estimate the action-value function, $Q(s,a;\theta) \approx Q^*(s,a)$, where $\theta$ are the parameters of the network. Exploration based on intrinsic rewards (e.g., Bellemare et al., 2016), which uses a DQN agent, additionally augments the observed rewards $r_t$ with a bonus $\beta/\sqrt{N(s_t, a_t)}$ based on

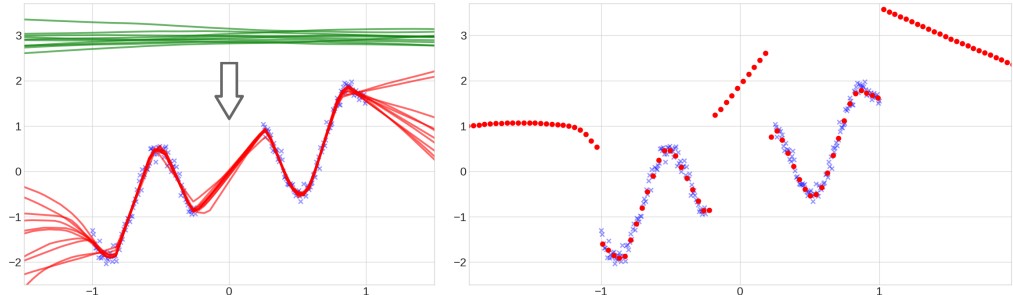

Figure 2: A simple regression task to illustrate the effect of an optimistic initialisation in neural networks. **Left:** 10 different networks whose final layer biases are initialised at 3 (shown in green), and the same networks after training on the blue data points (shown in red). **Right:** One of the trained networks whose output has been augmented with an optimistic bias as in equation 1. The counts were obtained by computing a histogram over the input space $[-2, 2]$ with 50 bins.

pseudo-visitation-counts $N(s_t, a_t)$. The DQN parameters $\theta$ are trained by gradient descent on the mean squared regression loss $\mathcal{L}$ with bootstrapped 'target' $y_t$:

$$\mathcal{L}[\theta] := \mathbb{E}\left[\left(\overbrace{r_t + \frac{\beta}{\sqrt{N(s_t, a_t)}} + \gamma \max_{a'} Q(s_{t+1}, a'; \theta^-)}^{y_t} - Q(s_t, a_t; \theta)\right)^2 \middle| (s_t, a_t, r_t, s_{t+1}) \sim D\right]. \quad (2)$$

The expectation is estimated with uniform samples from a replay buffer $D$ (Lin, 1992). $D$ stores past transitions $(s_t, a_t, r_t, s_{t+1})$, where the state $s_{t+1}$ is observed after taking the action $a_t$ in state $s_t$ and receiving reward $r_t$. To improve stability, DQN uses a target network, parameterised by $\theta^-$, which is periodically copied from the regular network and kept fixed for a number of iterations.

## 3 Optimistic Pessimistically Initialised $Q$-Learning

Our method Optimistic Pessimistically Initialised $Q$-Learning (OPIQ) ensures optimism in the $Q$-value estimates of unvisited, novel state-action pairs in order to drive exploration. This is achieved by augmenting the $Q$-value estimates in the following manner:

$$Q^+(s, a) := Q(s, a) + \frac{C}{(N(s, a) + 1)^M},$$

and using these $Q^+$-values during action selection and bootstrapping. In this section, we motivate OPIQ, analyse it in the tabular setting, and describe a deep RL implementation.

### 3.1 Motivations

**Optimistic initialisation does not work with neural networks.** For an optimistic initialisation to benefit exploration, the $Q$-values must start sufficiently high. More importantly, the values for unseen state-action pairs must *remain high*, until they are updated. When using a deep neural network to approximate the $Q$-values, we can initialise the network to output optimistic values, for example, by adjusting the final bias. However, after a small amount of training, the values for novel state-action pairs may not remain high. Furthermore, due to the generalisation of neural networks we cannot know how the values for these unseen state-action pairs compare to the trained state-action pairs. Figure 2 (left), which illustrates this effect for a simple regression task, shows that different initialisations can lead to dramatically different generalisations. It is therefore prohibitively difficult to use optimistic initialisation of a deep neural network to drive exploration.

**Instead, we augment our $Q$-value estimates with an optimistic bonus.** Our motivation for the form of the bonus in equation 1, $\frac{C}{(N(s,a)+1)^M}$, stems from UCB-H (Jin et al., 2018), where all tabular $Q$-values are initialised with $H$ and the first update for a state-action pair completely overwrites that value because the learning rate for the update ($\eta_1$) is 1. One can alternatively view these $Q$-values as zero-initialised with the additional term $Q(s, a) + H \cdot \mathbb{1}\{N(s, a) < 1\}$, where $N(s, a)$ is the visitation count for the state-action pair $(s, a)$. Our approach approximates the discrete indicator

---

**Algorithm 1** OPIQ algorithm

---

Initialise $Q_t(s,a) \leftarrow 0, N(s,a,t) \leftarrow 0, \forall(s,a,t) \in \mathcal{S} \times \mathcal{A} \times \{1,...,H,H+1\}$
**for** *each episode* $k = 1,...,K$ **do**
    **for** *each timestep* $t = 1,...,H$ **do**
        Take action $a_t \leftarrow \arg\max_a Q_t^+(s_t,a)$.
        Receive $r(s_t,a_t,t)$ and $s_{t+1}$.
        Increment $N(s_t,a_t,t)$.
        $Q_t(s_t,a_t) \leftarrow (1-\eta_N)Q_t(s_t,a_t) + \eta_N(r(s_t,a_t,t) + b_N^T + \min\{H, \max_{a'} Q_{t+1}^+(s_{t+1},a')\})$.
    **end**
**end**

---

function $\mathbb{1}$ as $(N(s,a)+1)^{-M}$ for sufficiently large $M$. However, since gradient descent cannot completely overwrite the $Q$-value estimate for a state-action pair after a single update, it is beneficial to have a smaller hyperparameter $M$ that governs how quickly the optimism decays.

**For a worst case analysis we assume all $Q$-value estimates are pessimistic.** In the common scenario where all rewards are nonnegative, the lowest possible return for an episode is zero. If we then zero-initialise our $Q$-value estimates, as is common for neural networks, we are starting with a *pessimistic* initialisation. As shown in Figure 2(left), we cannot predict how a neural network will generalise, and thus we cannot predict if the $Q$-value estimates for unvisited state-action pairs will be optimistic or pessimistic. We thus assume they are pessimistic in order to perform a worst case analysis. However, this is not a requirement: our method works with any initialisation and rewards.

In order to then approximate an optimistic initialisation, the scaling parameter $C$ in equation 1 can be chosen to guarantee unseen $Q^+$-values are overestimated, for example, $C := H$ in the undiscounted finite-horizon tabular setting and $C := 1/(1-\gamma)$ in the discounted episodic setting (assuming 1 is the maximum reward obtainable at each timestep). However, in some environments it may be beneficial to use a smaller parameter $C$ for faster convergence. These $Q^+$-values are then used both during action selection and during bootstrapping. Note that in the finite horizon setting the counts $N$, and thus $Q^+$, would depend on the timestep $t$.

**Hence, we split the optimistic $Q^+$-values into two parts: a pessimistic $Q$-value component and an optimistic component based solely on the counts for a state-action pair.** This separates our source of optimism from the neural network function approximator, yielding $Q^+$-values that remain high for unvisited state-action pairs, assuming a suitable counting scheme. Figure 2 (right) shows the effects of adding this optimistic component to a network's outputs.

**Optimistic $Q^+$-values provide an increased incentive to explore.** By using optimistic $Q^+$ estimates, especially during action selection and bootstrapping, the agent is incentivised to try and visit novel state-action pairs. Being optimistic during action selection in particular encourages the agent to try novel actions that have not yet been visitied. Without an optimistic estimate for novel state-action pairs the agent would have no incentive to try an action it has never taken before at a given state. Being optimistic during bootstrapping ensures the agent is incentivised to return to states in which it has not yet tried every action. This is because the maximum $Q^+$-value will be large due to the optimism bonus. Both of these effects lead to a strong incentive to explore novel state-action pairs.

## 3.2 TABULAR REINFORCEMENT LEARNING

In order to ensure that OPIQ has a strong theoretical foundation, we must ensure it is provably efficient in the tabular domain. We restrict our analysis to the finite horizon tabular setting and only consider building upon UCB-H (Jin et al., 2018) for simplicity. Achieving a better regret bound using UCB-B (Jin et al., 2018) and extending the analysis to the infinite horizon discounted setting (Dong et al., 2019) are steps for future work.

Our algorithm removes the optimistic initialisation of UCB-H, instead using a pessimistic initialisation (all $Q$-values start at 0). We then use our $Q^+$-values during action selection and bootstrapping. Pseudocode is presented in Algorithm 1.

**Theorem 1.** *For any $p \in (0, 1)$ , with probability at least $1 - p$ the total regret of $Q^+$ is at most $\mathcal{O}(\sqrt{H^4 SAT \log(SAT/p)})$ for $M \geq 1$ and at most $\mathcal{O}(H^{1+M} SAT^{1-M} + \sqrt{H^4 SAT \log(SAT/p)})$ for $0 < M < 1$.*

The proof is based on that of Theorem 1 from (Jin et al., 2018). Our $Q^+$-values are always greater than or equal to the $Q$-values that UCB-H would estimate, thus ensuring that our estimates are also greater than or equal to $Q^*$. Our overestimation relative to UCB-H is then governed by the quantity $H/(N(s, a) + 1)^M$, which when summed over all timesteps does not depend on $T$ for $M > 1$. As $M \to \infty$ we exactly recover UCB-H, and match the asymptotic performance of UCB-H for $M \geq 1$. Smaller values of $M$ result in our optimism decaying more slowly, which results in more exploration. The full proof is included in Appendix I.

We also show that OPIQ without optimistic action selection or the count-based intrinsic motivation term $b_N^T$ is not provably efficient by showing it can incur linear regret with high probability on simple MDPs (see Appendices G and H).

Our primary motivation for considering a tabular algorithm that pessimistically initialises its $Q$-values, is to provide a firm theoretical foundation on which to base a deep RL algorithm, which we describe in the next section.

### 3.3 DEEP REINFORCEMENT LEARNING

For deep RL, we base OPIQ on DQN (Mnih et al., 2015), which uses a deep neural network with parameters $\theta$ as a function approximator $Q_\theta$. During action selection, we use our $Q^+$-values to determine the greedy action:

$$a_t = \arg\max_a \left\{ Q_\theta(s, a) + \frac{C_{\text{action}}}{(N(s, a) + 1)^M} \right\}, \quad (3)$$

where $C_{\text{action}}$ is a hyperparameter governing the scale of the optimistic bias during action selection. In practice, we use an $\epsilon$-greedy policy. After every timestep, we sample a batch of experiences from our experience replay buffer, and use $n$-step $Q$-learning (Mnih et al., 2016). We recompute the counts for each relevant state-action pair, to avoid using stale pseudo-rewards. The network is trained by gradient decent on the loss in equation 2 with the target:

$$y_t := \sum_{i=0}^{n-1} \gamma^i \left( r(s_{t+i}, a_{t+i}) + \frac{\beta}{\sqrt{N(s_{t+i}, a_{t+i})}} \right) + \gamma^n \max_{a'} \left\{ Q_{\theta^-}(s_{t+n}, a') + \frac{C_{\text{bootstrap}}}{(N(s_{t+n}, a') + 1)^M} \right\}. \quad (4)$$

where $C_{\text{bootstrap}}$ is a hyperparameter that governs the scale of the optimistic bias during bootstrapping.

For our final experiments on Montezuma's Revenge we additionally use the Mixed Monte Carlo (MMC) target (Bellemare et al., 2016; Ostrovski et al., 2017), which mixes the target with the environmental monte carlo return for that episode. Further details are included in Appendix D.4.

We use the method of static hashing (Tang et al., 2017) to obtain our pseudocounts on the first 2 of 3 environments we test on. For our experiments on Montezuma's Revenge we count over a downsampled image of the current game frame. More details can be found in Appendix B.

A DQN with pseudocount derived intrinsic reward (DQN + PC) (Bellemare et al., 2016) can be seen as a naive extension of UCB-H to the deep RL setting. However, it does not attempt to ensure optimism in the $Q$-values used during action selection and bootstrapping, which is a crucial component of UCB-H. Furthermore, even if the $Q$-values were initialised optimistically at the start of training they would not remain optimistic long enough to drive exploration, due to the use of neural networks. OPIQ, on the other hand, is designed with these limitations of neural networks in mind. By augmenting the neural network's $Q$-value estimates with optimistic bonuses of the form $\frac{C}{(N(s,a)+1)^M}$, OPIQ ensures that the $Q^+$-values used during action selection and bootstrapping are optimistic. We can thus consider OPIQ as a *deep* version of UCB-H. Our results show that optimism during action selection and bootstrapping is extremely important for ensuring efficient exploration.

## 4 RELATED WORK

**Tabular Domain:** There is a wealth of literature related to provably efficient exploration in the tabular domain. Popular model-based algorithms such as R-MAX (Brafman and Tennenholtz, 2002),

MBIE (and MBIE-EB) (Strehl and Littman, 2008), UCRL2 (Jaksch et al., 2010) and UCBVI (Azar et al., 2017) are all based on the principle of *optimism in the face of uncertainty*. Osband and Van Roy (2017) adopt a Bayesian viewpoint and argue that posterior sampling (PSRL) (Strens, 2000) is more practically efficient than approaches that are optimistic in the face of uncertainty, and prove that in Bayesian expectation PSRL matches the performance of *any* optimistic algorithm up to constant factors. Agrawal and Jia (2017) prove that an optimistic variant of PSRL is provably efficient under a frequentist regret bound.

The only provably efficient model-free algorithms to date are delayed $Q$-learning (Strehl et al., 2006) and UCB-H (and UCB-B) (Jin et al., 2018). Delayed $Q$-learning optimistically initialises the $Q$-values that are carefully controlled when they are updated. UCB-H and UCB-B also optimistically initialise the $Q$-values, but also utilise a count-based intrinsic motivation term and a special learning rate to achieve a favourable regret bound compared to model-based algorithms. In contrast, OPIQ pessimistically initialises the $Q$-values. Whilst we base our current analysis on UCB-H, the idea of augmenting pessimistically initialised $Q$-values can be applied to any model-free algorithm.

**Deep RL Setting:** A popular approach to improving exploration in deep RL is to utilise intrinsic motivation (Oudeyer and Kaplan, 2009), which computes a quantity to add to the environmental reward. Most relevant to our work is that of Bellemare et al. (2016), which takes inspiration from MBIE-EB (Strehl and Littman, 2008). Bellemare et al. (2016) utilise the number of times a state has been visited to compute the intrinsic reward. They outline a framework for obtaining approximate counts, dubbed *pseudocounts*, through a learned density model over the state space. Ostrovski et al. (2017) extend the work to utilise a more expressive PixelCNN (van den Oord et al., 2016) as the density model, whereas Fu et al. (2017) train a neural network as a discriminator to also recover a density model. Machado et al. (2018a) instead use the successor representation to obtain generalised counts. Choi et al. (2019) learn a feature space to count that focusses on regions of the state space the agent can control, and Pathak et al. (2017) learn a similar feature space in order to provide the error of a learned model as intrinsic reward. A simpler and more generic approach to approximate counting is *static hashing* which projects the state into a lower dimensional space before counting (Tang et al., 2017). None of these approaches attempt to augment or modify the $Q$-values used for action-selection or bootstrapping, and hence do not attempt to ensure optimistic values for novel state-action pairs.

Chen et al. (2017) build upon bootstrapped DQN (Osband et al., 2016) to obtain uncertainty estimates over the $Q$-values for a given state in order to act optimistically by choosing the action with the largest UCB. However, they do not utilise optimistic estimates during bootstrapping. Osband et al. (2018) also extend bootstrapped DQN to include a prior by extending RLSVI (Osband et al., 2017) to deep RL. Osband et al. (2017) show that RLSVI achieves provably efficient Bayesian expected regret, which requires a prior distribution over MDPs, whereas OPIQ achieves provably efficient worse case regret. Bootstrapped DQN with a prior is thus a model-free algorithm that has strong theoretical support in the tabular setting. Empirically, however, its performance on sparse reward tasks is worse than DQN with pseudocounts.

Machado et al. (2015) shift and scale the rewards so that a zero-initialisation is optimistic. When applied to neural networks this approach does not result in optimistic $Q$-values due to the generalisation of the networks. Bellemare et al. (2016) empirically show that using a pseudocount intrinsic motivation term performs much better empirically on hard exploration tasks.

Choshen et al. (2018) attempt to generalise the notion of a count to include information about the counts of future state-actions pairs in a trajectory, which they use to provide bonuses during action selection. Oh and Iyengar (2018) extend delayed $Q$-learning to utilise these generalised counts and prove the scheme is PAC-MDP. The generalised counts are obtained through $E$-values which are learnt using SARSA with a constant 0 reward and $E$-value estimates initialised at 1. When scaling to the deep RL setting, these $E$-values are estimated using neural networks that cannot maintain their initialisation for unvisited state-action pairs, which is crucial for providing an incentive to explore. By contrast, OPIQ uses a separate source to generate the optimism necessary to explore the environment.

## 5 EXPERIMENTAL SETUP

We compare OPIQ against baselines and ablations on three sparse reward environments. The first is a randomized version of the Chain environment proposed by Osband et al. (2016) and used in (Shyam et al., 2019) with a chain of length 100, which we call Randomised Chain. The second is a two-dimensional maze in which the agent starts in the top left corner (white dot) and is only rewarded upon finding the goal (light grey dot). We use an image of the maze as input and randomise the actions similarly to the chain. The third is Montezuma's Revenge from Arcade Learning environment (Bellemare et al., 2013), a notoriously difficult sparse reward environment commonly used as a benchmark to evaluate the performance and scaling of Deep RL exploration algorithms.

See Appendix D for further details on the environments, baselines and hyperparameters used.[1]

### 5.1 ABLATIONS AND BASELINES

We compare OPIQ against a variety of DQN-based approaches that use pseudocount intrinsic rewards, the DORA agent (Choshen et al., 2018) (which generates count-like optimism bonuses using a neural network), and two strong exploration baselines:

$\epsilon$**-greedy DQN:** a standard DQN that uses an $\epsilon$-greedy policy to encourage exploration. We anneal $\epsilon$ linearly over a fixed number of timesteps from 1 to 0.01.
**DQN + PC:** we add an intrinsic reward of $\beta/\sqrt{N(s,a)}$ to the environmental reward based on (Bellemare et al., 2016; Tang et al., 2017).
**DQN R-Subtract (+PC):** we subtract a constant from all environmental rewards received when training, so that a zero-initialisation is optimistic, as described for a DQN in (Bellemare et al., 2016) and based on Machado et al. (2015).
**DQN Bias (+PC):** we initialise the bias of the final layer of the DQN to a positive value at the start of training as a simple method for optimistic initialisation with neural networks.
**DQN + DORA:** we use the *generalised counts* from (Choshen et al., 2018) as an intrinsic reward.
**DQN + DORA OA:** we additionally use the generalised counts to provide an optimistic bonus during action selection.
**DQN + RND:** we add the RND bonus from (Burda et al., 2018) as an intrinsic reward.
**BSP:** we use Bootstrapped DQN with randomised prior functions (Osband et al., 2018).

In order to better understand the importance of each component of our method, we also evaluate the following ablations:

**Optimistic Action Selection (OPIQ w/o OB):** we only use our $Q^+$-values during action selection, and use $Q$ during bootstrapping (without Optimistic Bootstrapping). The intrinsic motivation term remains.
**Optimistic Action Selection and Bootstrapping (OPIQ w/o PC):** we use our $Q^+$-values during action selection and bootstrapping, but do not include an intrinsic motivation term (without Pseudo Counts).

## 6 RESULTS

### 6.1 RANDOMISED CHAIN

We first consider the visually simple domain of the randomised chain and compare the count-based methods. Figure 3 shows the performance of OPIQ compared to the baselines and ablations. OPIQ significantly outperforms the baselines, which do not have any explicit mechanism for optimism during action selection. A DQN with pseudocount derived intrinsic rewards is unable to reliably find the goal state, but setting the final layer's bias to one produces much better performance. For the DQN variant in which a constant is subtracted from all rewards, all of the configurations (including those with pseudocount derived intrinsic bonuses) were unable to find the goal on the right and thus the agents learn quickly to latch on the inferior reward of moving left.

Compared to its ablations, OPIQ is more stable in this task. OPIQ without pseudocounts performs similarly to OPIQ but is more varied across seeds, whereas the lack of optimistic bootstrapping results in worse performance and significantly more variance across seeds.

---

[1]Code is available at: https://github.com/oxwhirl/opiq.

## 6.2 MAZE

We next consider the harder and more visually complex task of the Maze and compare against all baselines.

Figure 4 shows that only OPIQ is able to find the goal in the sparse reward maze. This indicates that explicitly ensuring optimism during action selection and bootstrapping can have a significant positive impact in sparse reward tasks, and that a naive extension of UCB-H to the deep RL setting (DQN + PC) results in insufficient exploration.

Figure 4 (right) shows that attempting to ensure optimistic $Q$-values by adjusting the bias of the final layer (DQN Bias + PC), or by subtracting a constant from the reward (DQN R-Subtract + PC) has very little effect.

As expected DQN + RND performs poorly on this domain compared to the pseudocount based methods. The visual input does not vary much across the state space, resulting in the RND bonus failing to provide enough intrinsic motivation to ensure efficient exploration. Additionally it does not feature any explicit mechanism for optimism during action selection, and thus Figure 4 (right) shows it explores the environment relatively slowly.

Both DQN+DORA and DQN+DORA OA also perform poorly in this domain since their source of intrinsic motivation disappears quickly. As noted in Figure 2, neural networks do not maintain their starting initialisations after training. Thus, the intrinsic reward DORA produces goes to 0 quickly since the network producing its bonuses learns to generalise quickly.

BSP is the only exploration baseline we test that does not add an intrinsic reward to the environmental reward, and thus it performs poorly compared to the other baselines on this environment.

Figure 5 shows that OPIQ and all its ablations manage to find the goal in the maze. OPIQ also explores slightly faster than its ablations (right), which shows the benefits of optimism during both action selection and bootstrapping. In addition, the episodic reward for the the ablation without optimistic bootstrapping is noticeably more unstable (Figure 5, left). Interestingly, OPIQ without pseudocounts performs significantly worse than the other ablations. This is surprising since the theory suggests that the count-based intrinsic motivation is only required when the reward or transitions of the MDP are stochastic (Jin et al., 2018), which is not the case here. We hypothesise that adding PC-derived intrinsic bonuses to the reward provides an easier learning problem, especially when using $n$-step $Q$-Learning, which yields the performance gap.

However, our results show that the PC-derived intrinsic bonuses are not enough on their own to ensure sufficient exploration. The large difference in performance between DQN + PC and OPIQ w/o OB is important, since they only differ in the use of optimistic action selection. The results in Figures 4 and 5 show that optimism during action selection is extremely important in exploring the environment efficiently. Intuitively, this makes sense, since this provides an incentive for the agent to try actions it has never tried before, which is crucial in exploration.

Figure 6 visualises the values used during action selection for a DQN + PC agent and OPIQ, showing the count-based augmentation provides optimism for relatively novel state-action pairs, driving the agent to explore more of the state-action space.

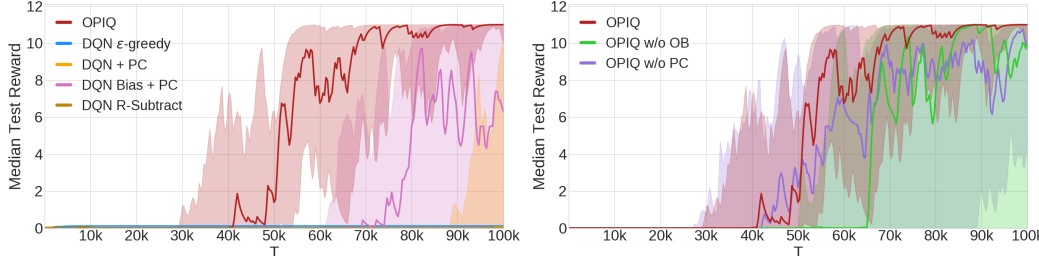

Figure 3: Results for the randomised chain environment. Median across 20 seeds is plotted and the 25%-75% quartile is shown shaded. **Left:** OPIQ outperforms the baselines. **Right:** OPIQ is more stable than its ablations.

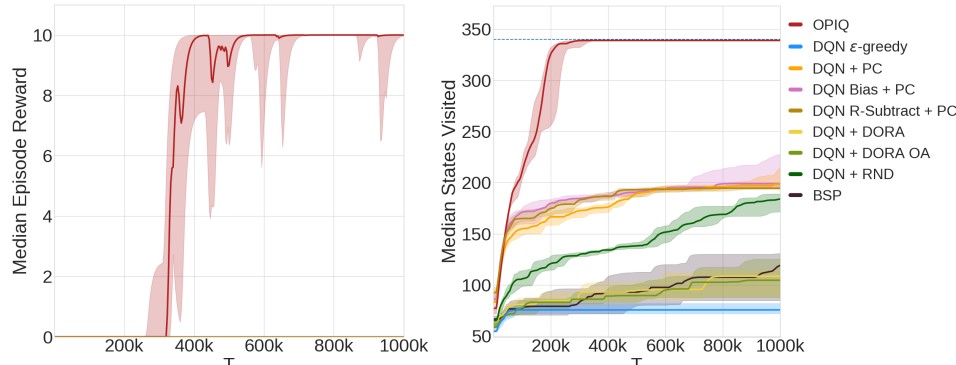

Figure 4: Results for the maze environment comparing OPIQ and baselines. Median across 8 seeds is plotted and the 25%-75% quartile is shown shaded. **Left:** The episode reward. **Right:** Number of distinct states visited over training. The total number of states in the environment is shown as a dotted line.

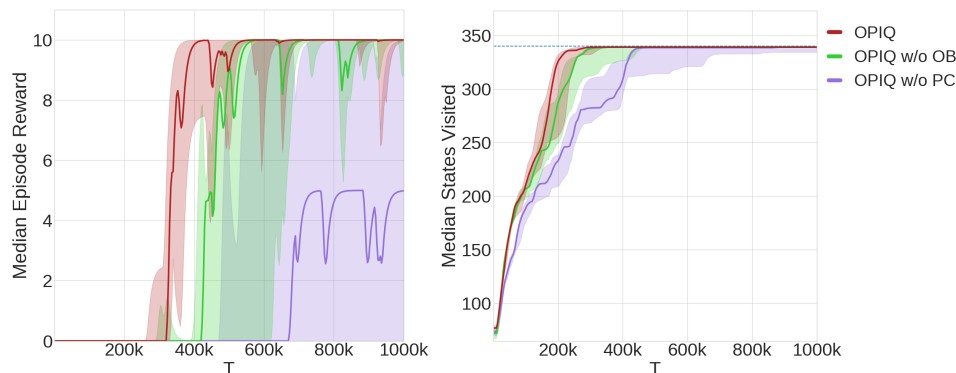

Figure 5: Results for the maze environment comparing OPIQ and ablations. Median across 8 seeds is plotted and the 25%-75% quartile is shown shaded. **Left:** The episode reward. **Right:** Number of distinct states visited over training. The total number of states in the environment is shown as a dotted line.

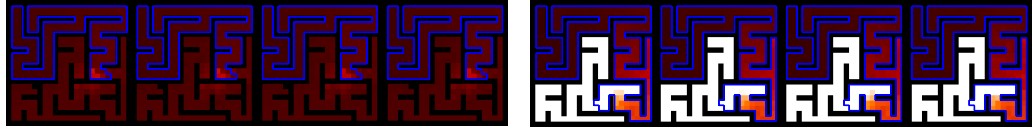

Figure 6: Values used during action selection for each of the 4 actions. The region in blue indicates states that have already been visited. Other colours denote $Q$-values between 0 (black) and 10 (white). **Left:** The $Q$-values used by DQN with pseudocounts. **Right:** $Q^+$-values used by OPIQ with $C_{\text{action}} = 100$.

## 6.3 MONTEZUMA'S REVENGE

Finally, we consider Montezuma's Revenge, one of the hardest sparse reward games from the ALE (Bellemare et al., 2013). Note that we only train up to 12.5mil timesteps (50mil frames), a $1/4$ of the usual training time (50mil timesteps, 200mil frames).

Figure 7 shows that OPIQ significantly outperforms the baselines in terms of the episodic reward and the maximum episodic reward achieved during training. The higher episode reward and much higher maximum episode reward of OPIQ compared to DQN + PC once again demonstrates the importance of optimism during action selection and bootstrapping.

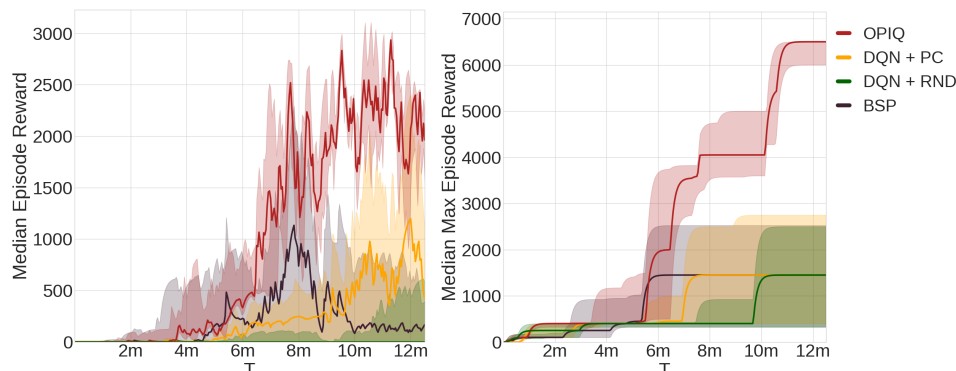

Figure 7: Results for Montezuma's Revenge. Median across 4 seeds is plotted and the 25%-75% quartile is shown shaded. **Left:** The episode reward. **Right:** The maximum reward achieved during an episode.

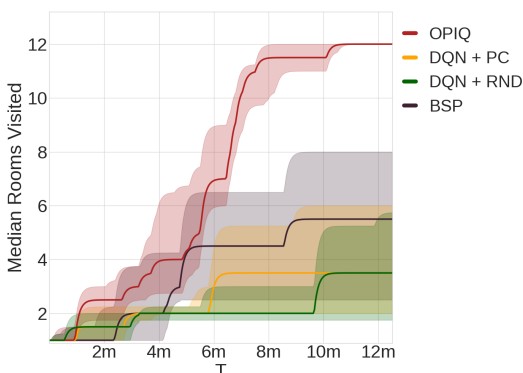

Figure 8: Further Results for Montezuma's Revenge showing the number of rooms visited over training comparing OPIQ and baselines. Median across 4 seeds is plotted and the 25%-75% quartile is shown shaded.

In this environment BSP performs much better than in the Maze, but achieves significantly lower episodic rewards than OPIQ.

Figure 8 shows the distinct number of rooms visited across the training period. We can see that OPIQ manages to reliably explore 12 rooms during the 12.5mil timesteps, significantly more than the other methods, thus demonstrating its improved exploration in this complex environment.

Our results on this challenging environment show that OPIQ can scale to high dimensional complex environments and continue to provide significant performance improvements over an agent only using pseudocount based intrinsic rewards.

# 7 CONCLUSIONS AND FUTURE WORK

This paper presented OPIQ, a model-free algorithm that does not rely on an optimistic initialisation to ensure efficient exploration. Instead, OPIQ augments the $Q$-values estimates with a count-based optimism bonus. We showed that this is provably efficient in the tabular setting by modifying UCB-H to use a pessimistic initialisation and our augmented $Q^+$-values for action selection and bootstrapping. Since our method does not rely on a specific initialisation scheme, it easily scales to deep RL when paired with an appropriate counting scheme. Our results showed the benefits of maintaining optimism both during action selection and bootstrapping for exploration on a number of hard sparse reward environments including Montezuma's Revenge. In future work, we aim to extend OPIQ by integrating it with more expressive counting schemes.

## 8 ACKNOWLEDGEMENTS

We would like to thank the entire WhiRL lab for their helpful feedback, in particular Gregory Farquhar and Supratik Paul. We would also like to thank the anonymous reviewers for their constructive comments during the reviewing process. This project has received funding from the European Research Council (ERC), under the European Union's Horizon 2020 research and innovation programme (grant agreement number 637713). It was also supported by an EPSRC grant (EP/M508111/1, EP/N509711/1). The experiments were made possible by a generous equipment grant from NVIDIA and the JP Morgan Chase Faculty Research Award.

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

## A  BACKGROUND

### A.1  TABULAR REINFORCEMENT LEARNING

For the tabular setting, we consider a discrete finite-horizon Markov Decision Process (MDP), which can be defined as a tuple $(\mathcal{S}, \mathcal{A}, \{P_t\}, \{R_t\}, H, \rho)$, where $\mathcal{S}$ is the finite state space, $\mathcal{A}$ is the finite action space, $P_t(\cdot|s, a)$ is the state-transition distribution for timestep $t = 1, ..., H$, $R_t(\cdot|s, a)$ is the distribution over rewards after taking action $a$ in state $s$, $H$ is the horizon, and $\rho$ is the distribution over starting states. Without loss of generality we assume that $R_t(\cdot|s, a) \in [0, 1]$. We use $S$ and $A$ to denote the number of states and the number of actions, respectively, and $N(s, a, t)$ as the number of times a state-action pair $(s, a)$ has been visited at timestep $t$.

Our goal is to find a set of policies $\pi_t : \mathcal{S} \to \mathcal{A}$, $\pi := \{\pi_t\}$, that chooses the agent's actions at time $t$ such that the expected sum of future rewards is maximised. To this end we define the $Q$-value at time $t$ of a given policy $\pi$ as $Q_t^\pi(s, a) := \mathbb{E}\left[r + Q_{t+1}^\pi(s', \pi_{t+1}(s')) \mid r \sim R_t(\cdot|s, a), s' \sim P_t(\cdot|s, a)\right]$, where $Q_t^\pi(s, a) = 0, \forall t > H$. The agent interacts with the environment for $K$ episodes, $T := KH$, yielding a total regret: $\text{Regret}(K) = \sum_{k=1}^K \left(\max_{\pi^*} Q_1^{\pi^*}(s_1^k, \pi_1^*(s_1^k)) - Q_1^{\pi^k}(s_1^k, \pi_1^k(s_1^k))\right)$. Here $s_1^k$ refers to the starting state and $\pi^k$ to the policy at the beginning of episode $k$. We are interested in bounding the worst case total regret with probability $1 - p, 0 < p < 1$.

UCB-H (Jin et al., 2018) is an online $Q$-learning algorithm for the finite-horizon setting outlined above where the worse case total regret is bounded with a probability of $1 - p$ by $\mathcal{O}(\sqrt{H^4 SAT \log(SAT/p)})$. All $Q$-values for timesteps $t \le H$ are optimistically initialised at $H$. The learning rate is defined as $\eta_N = \frac{H+1}{H+N}$, where $N := N(s_t, a_t, t)$ is the number of times state-action pair $(s_t, a_t)$ has been observed at step $t$ and $\eta_1 = 1$ at the first encounter of any state-action pair. The update rule for a transition at step $t$ from state $s_t$ to $s_{t+1}$, after executing action $a_t$ and receiving reward $r_t$, is:

$$Q_t(s_t, a_t) \leftarrow (1 - \eta_N) Q_t(s_t, a_t) + \eta_N (r_t + b_N^T + \min\{H, \max_{a'} Q_{t+1}(s_{t+1}, a')\}), \quad (5)$$

where $b_N^T := 2\sqrt{\frac{H^3 \log(SAT/p)}{N}}$, is the count-based intrinsic motivation term.[2]

## B  COUNTING IN LARGE, COMPLEX STATE SPACES

In deep RL, the primary difficulty for exploration based on count-based intrinsic rewards is obtaining appropriate state-action counts. In this paper we utilise approximate counting schemes (Bellemare et al., 2016; Ostrovski et al., 2017; Tang et al., 2017) in order to cope with continuous and/or high-dimensional state spaces. In particular, for the chain and maze environments we use *static hashing* (Tang et al., 2017), which projects a state $s$ to a low-dimensional feature vector $\phi(s) = \text{sign}(Af(s))$, where $f$ flattens the state $s$ into a single dimension of length $D$; $A$ is a $k \times D$ matrix whose entries are initialised i.i.d. from a unit Gaussian: $\mathcal{N}(0, 1)$; and $k$ is a hyperparameter controling the *granularity* of counting: higher $k$ leads to more distinguishable states at the expense of generalisation.

Given the vector $\phi(s)$, we use a counting bloom filter (Fan et al., 2000) to update and retrieve its counts efficiently. To obtain counts $N(s, a)$ for state-action pairs, we maintain a separate data structure of counts for each action (the same vector $\phi(s)$ is used for all actions). This counting scheme is tabular and hence the counts for sufficiently different states do not interfere with one another. This ensures $Q^+$-values for unseen state-action pairs in equation 1 are large.

For our experiments on Montezuma's Revenge we use the same method of downsampling as in (Ecoffet et al., 2019), in which the greyscale state representation is resized from (42x42) to (11x8) and then binned from $\{0, ..., 255\}$ into 8 categories. We then maintain tabular counts over the new representation.

## C  GRANULARITY OF THE COUNTING MODEL

The granularity of the counting scheme is an important modelling consideration. If it is too granular, then it will assign an optimistic bias in regions of the state space where the network should be trusted

---

[2] Jin et al. (2018) use $\exists c > 0$ s.t. $b_N^T := c\sqrt{\frac{H^3 \log(SAT/p)}{N}}$ in their proof, but we find that $c = 2$ suffices.

to generalise. On the other hand, if it is too coarse then it could fail to provide enough of an optimistic bias in parts of the state space where exploration is still required. Figures 9 shows the differences between 2 levels of granularity. Taïga et al. (2018) provide a much more detailed analysis on the granularity of the count-based model, and its implications on the learned $Q$-values.

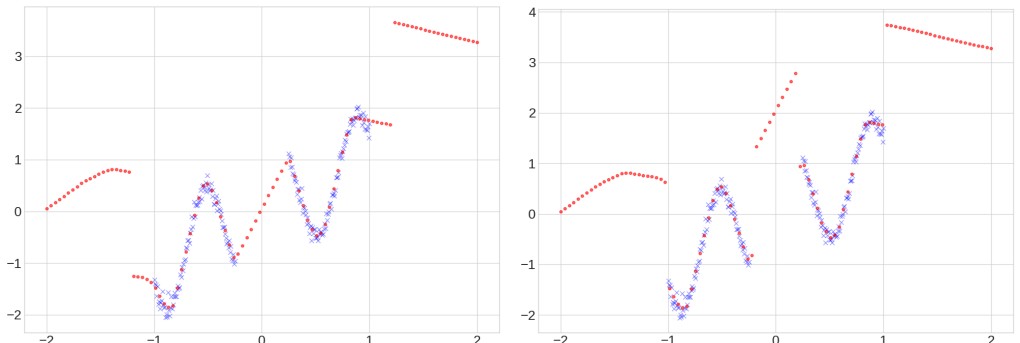

Figure 9: We consider the counting scheme from Figure 2, but vary the number of bins used. **Left:** 6 bins are used. Only the data points far from the training data are given an optimistic bias. **Right:** 50 bins are used. An optimistic bias is given to all data points that are not very close to the training data.

# D    EXPERIMENTAL SETUP

## D.1    ENVIRONMENTS

### D.1.1    RANDOMISED CHAIN

A randomized version of the Chain environment proposed by Osband et al. (2016) and used in (Shyam et al., 2019). We use a Chain of length 100. The agent starts the episode in State 2, and interacts with the MDP for 109 steps, after which the agent is reset. The agent has 2 actions that can move it Left or Right. At the beginning of training the action which takes the agent left or right at each state is randomly picked and then fixed. The agent receives a reward of $0.001$ for going Left in State 1, a reward of 1 for going Right in State 100 and no reward otherwise. The optimal policy is thus to pick the action that takes it Right at each timestep. Figure 10 shows the structure of the 100 Chain. Similarly to Osband et al. (2016) we use a thermometer encoding for the state: $\phi(s) := (\mathbb{1}\{x \le s\}) \in \{0,1\}^100$.

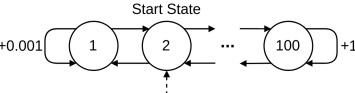

Figure 10: 100 Chain environment.

### D.1.2    MAZE

A 2-dimensional gridworld maze with a sparse reward in which the agent can move Up, Down, Left or Right. The agent starts each episode at a fixed location and must traverse through the maze in order to find the goal which provides $+10$ reward and terminates the episode, all other rewards are 0. The agent interacts with the maze for 250 timesteps before being reset. Empty space is represented by a 0, walls are 1, the goal is 2 and the player is 3. The state representation is a greyscaled image of the entire grid where each entry is divided by 3 to lie in $[0,1]$. The shape of the state representation is: $(24, 24, 1)$. Once again the effect of each action is randomised at each state at the beginning of training. Figure 11 shows the structure of the maze environment.

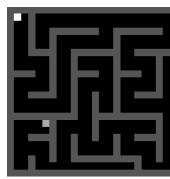

Figure 11: Maze environment.

### D.1.3 MONTEZUMA'S REVENGE

We follow the suggestions in (Machado et al., 2018b) and use the same environmental setup as used in (Burda et al., 2018). Specifically, we use stick actions with a probability of $p = 0.25$, a frame skip of 4 and do not show a terminal state on the loss of life.

## D.2 HYPERPARAMETERS AND NETWORK ARCHITECTURES

In all experiments we set $\gamma = 0.99$, use RMSProp with a learning rate of $0.0005$ and scale the gradient norms during training to be at most 5.

### D.2.1 RANDOMISED CHAIN

The network used is a MLP with 2 hidden layers of 256 units and ReLU non-linearities. We use 1 step $Q$-Learning.

Training lasts for 100k timesteps. $\epsilon$ is fixed at 0.01 for all methods except for $\epsilon$-greedy DQN in which it is linearly decayed from 1 to 0.01 over $\{100, 50k, 100k\}$ timesteps. We train on a batch size of 64 after every timestep with a replay buffer of size 10k. The target network is updated every 200 timesteps. The embedding size used for the counts is 32. We set $\beta = 0.1$ for the scale of the count-based intrinsic motivation.

For reward subtraction we consider subtracting $\{0.1, 1, 10\}$ from the reward. For an optimistic initialisation bias, we consider setting the final layer's bias to $\{0.1, 1, 10\}$. We consider both of the methods with and without count-based intrinsic motivation.

For OPIQ and its ablations we consider: $M \in \{0.1, 0.5, 2, 10\}, C_{\text{action}} \in \{0.1, 1, 10\}, C_{\text{bootstrap}} \in \{0.01, 0.1, 1, 10\}$.

For all methods we run 20 independent runs across the cross-product of all relevant parameters considered. We then sort them by the median test reward (largest area underneath the line) and report the median, lower and upper quartiles.

The best hyperparameters we found were:

**DQN $\epsilon$-greedy:** Decay rate: 100 timesteps.

**Optimistic Initialisation Bias:** Bias: 1, Pseudocount intrinsic motivation: True.

**Reward Subtraction:** Constant to subtract: 1, Pseudocount intrinsic motivation: False.

**OPIQ:** M: 0.5, $C_{\text{action}}$: 1, $C_{\text{bootstrap}}$: 1.

**OPIQ without Optimistic Bootstrapping:** M: 2, $C_{\text{action}}$: 10.

**OPIQ without Pseudocounts:** M: 2, $C_{\text{action}}$: 10, $C_{\text{bootstrap}}$: 10.

For Figure 13 the best hyperparameters for OPIQ with differing values of $M$ are:

**M: 0.1:** $C_{\text{action}}$: 10, $C_{\text{bootstrap}}$: 1.
**M: 0.5:** $C_{\text{action}}$: 1, $C_{\text{bootstrap}}$: 1.
**M: 2:** $C_{\text{action}}$: 10, $C_{\text{bootstrap}}$: 1.
**M: 10:** $C_{\text{action}}$: 10, $C_{\text{bootstrap}}$: 10.

### D.2.2 MAZE

The network used is the following feedforward network:

(State input: (24,24,1))
$\rightarrow$ (Conv Layer, 3x3 Filter, 16 Channels, Stride 2) $\rightarrow$ ReLU
$\rightarrow$ (Conv Layer, 3x3 Filter, 16 Channels, Stride 2) $\rightarrow$ ReLU
$\rightarrow$ Flatten
$\rightarrow$ (FC Layer, 400 Units) $\rightarrow$ ReLU
$\rightarrow$ (FC Layer, 200 Units)
$\rightarrow$ $\mathcal{A} = 4$ outputs.

We use 3 step $Q$-Learning.

Training lasts for 1mil timesteps. $\epsilon$ is decayed linearly from 1 to 0.01 over 50k timesteps for all methods except for $\epsilon$-greedy DQN in which it is linearly decayed from 1 to 0.01 over $\{100, 50k, 100k\}$ timesteps. We train on a batch of 64 after every timestep with a replay buffer of size 250k. The target network is updated every 1000 timesteps. The embedding dimension for the counts is 128.

For DQN + PC we consider $\beta \in \{0.01, 0.1, 1, 10, 100\}$. For all other methods we set $\beta = 0.1$ as it performed best.

For reward subtraction we consider subtracting $\{0.1, 1, 10\}$ from the reward. For an optimistic initialisation bias, we consider setting the final layer's bias to $\{0.1, 1, 10\}$. Both methods utilise a count-based intrinsic motivation.

For OPIQ and its ablations we set $M = 2$ since it worked best in preliminary experiments. We consider: $C_{\text{action}} \in \{0.1, 1, 10, 100\}, C_{\text{bootstrap}} \in \{0.01, 0.1, 1, 10\}$.

For the RND bonus we use the same architecture as the DQN for both the target and predictor networks, except the output is of size 128 instead of $|\mathcal{A}|$. We scale the squared error by $\beta_{rnd} \in \{0.001, 0.01, 0.1, 1, 10, 100\}$:

For DQN + DORA we use the same architecture for the $E$-network as the DQN. We add a sigmoid non-linearity to the output and initialise the final layer's weights and bias to 0 as described in (Choshen et al., 2018). We sweep across the scale of the intrinsic reward $\beta_{dora} \in \{\}$. For DQN + DORA OA we use $\beta_{dora} =$ and sweep across $\beta_{dora\_action} \in \{\}$.

For BSP we use the following architecture:

Shared conv layers:
(State input: (24,24,1))
$\rightarrow$ (Conv Layer, 3x3 Filter, 16 Channels, Stride 2) $\rightarrow$ ReLU
$\rightarrow$ (Conv Layer, 3x3 Filter, 16 Channels, Stride 2) $\rightarrow$ ReLU
$\rightarrow$ Flatten

$Q$-value Heads:
$\rightarrow$ (FC Layer, 400 Units) $\rightarrow$ ReLU
$\rightarrow$ (FC Layer, 200 Units)
$\rightarrow$ $\mathcal{A} = 4$ outputs.

We use $K = 10$ different bootstrapped DQN heads, and sweep over $\beta_{bsp} \in \{0.1, 1, 3, 10, 30, 100\}$.

For all methods we run 8 independent runs across the cross-product of all relevant parameters considered. We then sort them by the median episodic reward (largest area underneath the line) and report the median, lower and upper quartiles.

The best hyperparameters we found were:

**DQN $\epsilon$-greedy:** Decay rate: 100k timesteps.

**DQN + PC:** $\beta = 0.1$.

**Optimistic Initialisation Bias:** Bias: 1.

**Reward Subtraction:** Constant to subtract: 0.1.

**DQN + RND:** $\beta_{rnd} = 10$.

**DQN + DORA:** $\beta_{dora} = 0.01$.

**DQN + DORA OA:** $\beta_{dora} = 0.01$ and $\beta_{dora\_action} = 0.1$.

**BSP:** $\beta_{bsp} = 100$.

**OPIQ:** M: 2, $C_{\text{action}}$: 100, $C_{\text{bootstrap}}$: 0.01.

**OPIQ without Optimistic Bootstrapping:** M: 2, $C_{\text{action}}$: 100.

**OPIQ without Pseudocounts:** M: 2, $C_{\text{action}}$: 100, $C_{\text{bootstrap}}$: 0.1.

### D.2.3 MONTEZUMA'S REVENGE

The network used is the standard DQN used for Atari (Mnih et al., 2015; Bellemare et al., 2016).

We use 3 step $Q$-Learning.

Training lasts for 12.5mil timesteps (50mil frames in Atari). $\epsilon$ is decayed linearly from 1 to 0.01 over 1mil timesteps. We train on a batch of 32 after every 4th timestep with a replay buffer of size 1mil. The target network is updated every 8000 timesteps.

For all methods we consider $\beta_{mmc} \in \{0.005, 0.01, 0.025\}$.

For DQN + PC we consider $\beta \in \{0.01, 0.1, 1\}$.

For OPIQ and its ablations we set $M = 2$. We consider: $C_{\text{action}} \in \{0.1, 1\}, C_{\text{bootstrap}} \in \{0.01, 0.1\}$, $\beta \in \{0.01, 0.1\}$.

For the RND bonus we use the same architectures as in (Burda et al., 2018) (target network is smaller than the learned predictor network) except we use ReLU non-linearity. The output is the same of size 512. We scale the squared error by $\beta_{rnd} \in \{0.001, 0.01, 0.1, 1\}$:

For BSP we use the same architecture as in (Osband et al., 2018).

We use $K = 10$ different bootstrapped DQN heads, and sweep over $\beta_{bsp} \in \{0.1, 1, 10, 100\}$.

For all methods we run 4 independent runs across the cross-product of all relevant parameters considered. We then sort them by the median maximum episodic reward (largest area underneath the line) and report the median, lower and upper quartiles.

The best hyperparameters we found were:

**DQN + PC:** $\beta = 0.01, \beta_{mmc} = 0.01$.

**DQN + RND:** $\beta_{rnd} = 0.1, \beta_{mmc} = 0.01$.

**BSP:** $\beta_{bsp} = 0.1, \beta_{mmc} = 0.025$.

**OPIQ:** M= 2, $C_{\text{action}} = 0.1, C_{\text{bootstrap}} = 0.01, \beta_{mmc} = 0.01$.

**OPIQ without Optimistic Bootstrapping:** M= 2, $C_{\text{action}} = 0.1, \beta_{mmc} = 0.005$.

### D.3 BASELINES TRAINING DETAILS

**DQN + RND**: We do a single gradient descent step on a minibatch of the 32 most recently visited states. We also recompute the intrinsic rewards when sampling minibatches to train the DQN. The intrinsic reward used for a state $s$, is the squared error between the predictor network and the target network $\beta_{rnd}||\text{predictor}(s) - \text{target}(s)||_2^2$.

**DQN + DORA**: We train the $E$-values network using $n$-step SARSA (same $n$ as the DQN) with $\gamma_E = 0.99$. We maintain a replay buffer of size (batch size $* 4$) and sample batch size elements to train every timestep. The intrinsic reward we use is $\frac{\beta_{dora}}{\sqrt{-\log E(s,a)}}$.

**DQN + DORA OA**: We train the DQN + DORA agent described above and additionally augment the $Q$-values used for action selection with $\frac{\beta_{dora\_action}}{\sqrt{-\log E(s,a)}}$.

**BSP**: We train each Bootstrapped DQN head on all of the data from the replay buffer (as is done in (Osband et al., 2016; 2018). We normalise the gradients of the shared part of the network by $1/K$, where $K$ is the number of heads. The output of each head is $Q_k + \beta_{bsp}p_k$, where $p_k$ is a randomly initialised network (of the same architecture as $Q_k$) which is kept fixed throughout training. $\beta_{bsp}$ is a hyperparameter governing the scale of the prior regularisation.

### D.4 MIXED MONTE CARLO RETURN

For our experiments on Montezuma's Revenge we additionally mixed the 3 step $Q$-Learning target with the environmental rewards monte carlo return for the episode.

That is, the 3 step targets $y_t$ become:

$$y_{mmc} := (1 - \beta_{mmc})y_t + \beta_{mmc}(\sum_{i=0}^{\infty} \gamma^i r(s_{t+i}, a_{t+i}))$$

If the episode hasn't finished yet, we used 0 for the monte carlo return.

Our implementation differs from (Bellemare et al., 2016; Ostrovski et al., 2017) in that we do not use the intrinsic rewards as part of the monte carlo return. This is because we recompute the intrinsic rewards whenever we are using them as part of the targets for training, and recomputing all the intrinsic rewards for an entire episode (which can be over 1000 timesteps) is computationally prohibitive.

## E FURTHER RESULTS

### E.1 RANDOMISED CHAIN

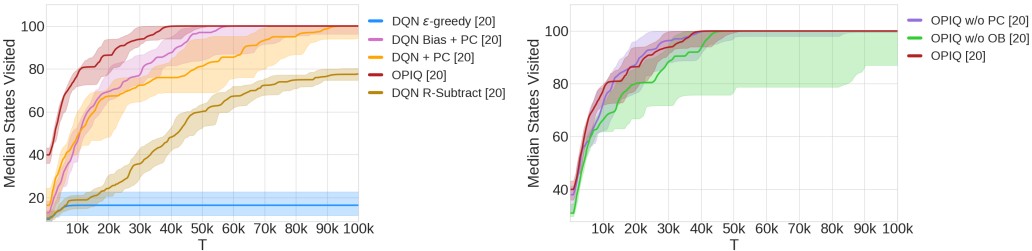

Figure 12: The number of distinct states visited over training for the chain environment. The median across 20 seeds is plotted and the 25%-75% quartile is shown shaded.

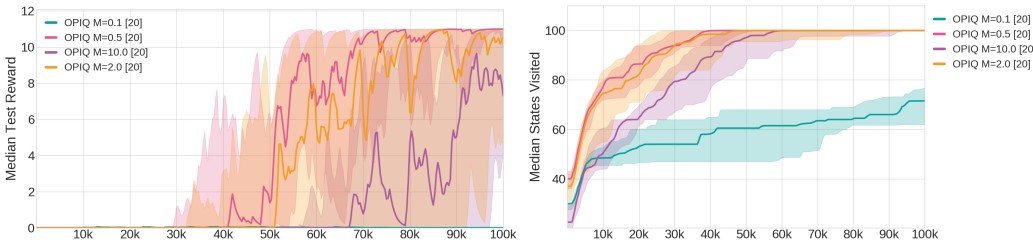

Figure 13: Comparing the performance of $M \in \{0.1, 0.5, 2, 10\}$ on the chain environment. The best hyperparameter combination for the differing values of $M$ is shown. The median across 20 seeds is plotted and the 25%-75% quartile is shown shaded.

We can see that OPIQ and ablations explore the environment much more quickly than the count-based baselines. The ablation without optimistic bootstrapping exhibits significantly more variance than the other ablations, showing the importance of optimism during bootstrapping. On this simple task the ablation without count-based intrinsic motivation performs on par with the full OPIQ. This is most

likely due to the simpler nature of the environment that makes propagating rewards much easier than the Maze. The importance of directed exploration is made abundantly clear by the $\epsilon$-greedy baseline that fails to explore much of the environment.

Figure 13 compares OPIQ with differing values of $M$. We can clearly see that a small value of $0.1$ results in insufficient exploration, due to the over-exploration of already visited state-action pairs. Additionally if $M$ is too large then the rate of exploration suffers due to the decreased optimism. On this task we found that $M = 0.5$ performed best, but on the harder Maze environment we found that $M = 2$ was better in preliminary experiments.

## E.2   MAZE

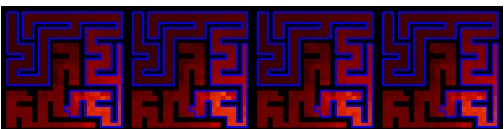

Figure 14: The $Q^+$-values OPIQ used during bootstrapping with $C_{\text{bootstrap}} = 0.01$.

Figure 14 shows the values used during bootstrapping for OPIQ. These $Q$-values show optimism near the novel state-action pairs which provides an incentive for the agent to return to this area of the state space.

## E.3   MONTEZUMA'S REVENGE

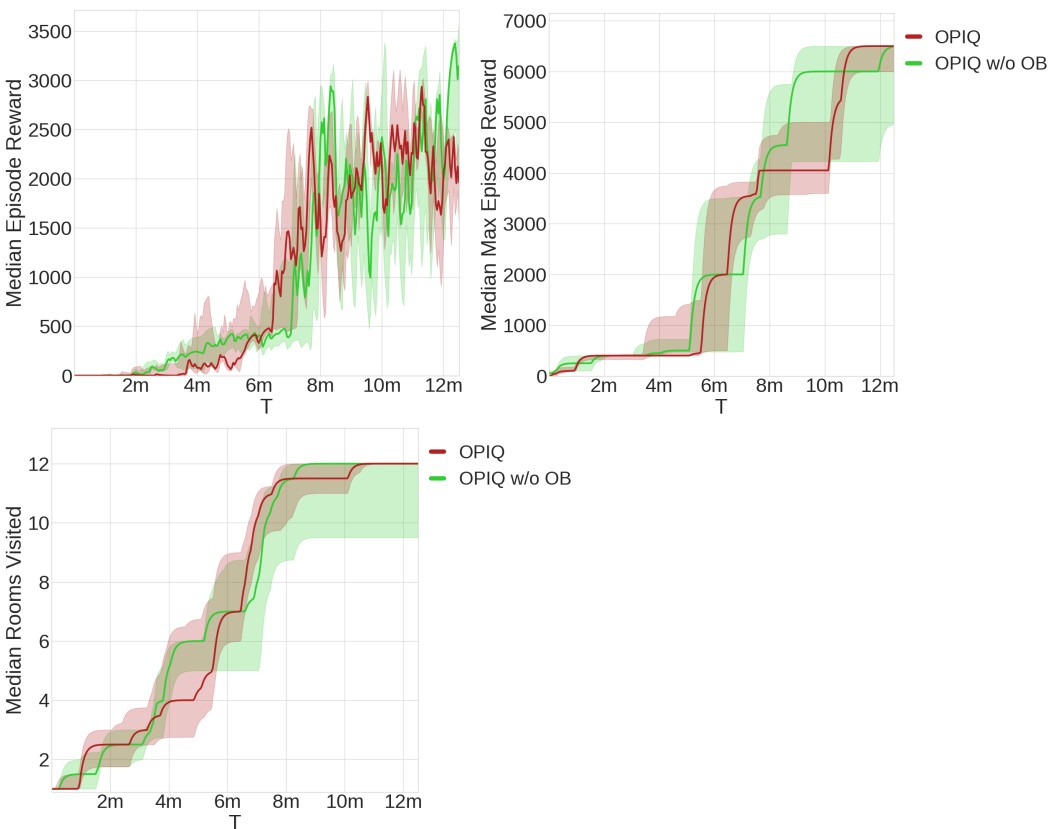

Figure 15: Results for Montezuma's Revenge comparing OPIQ and ablation. Median across 4 seeds is plotted and the 25%-75% quartile is shown shaded.

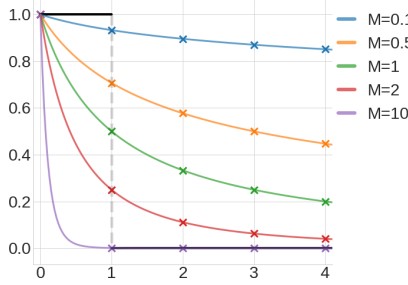

Figure 16: $\frac{1}{(x+1)^M}$ for various values of $M$, and the indicator function $\mathbb{1}\{x < 1\}$ shown in black. Higher values of $M$ provide a better approximation at integer values of $x$ (shown as crosses).

Figure 15 shows further results on Montezuma's Revenge comparing OPIQ and its ablation without optimistic bootstraping (OPIQ w/o OB). Similarly to the Chain and Maze results, we can see that OPIQ w/o OB performs similarly to the full OPIQ but has a higher variance across seeds.

## F MOTIVATIONS

Figure 16 compares various values of $M$ with the indicator function $\mathbb{1}\{x < 1\}$.

## G NECESSITY FOR OPTIMISM DURING ACTION SELECTION

To emphasise the necessity of optimistic $Q$-value estimates during exploration, we analyse the simple failure case for pessimistically initialised greedy $Q$-learning provided in the introduction. We use Algorithm 1, but use $Q$ instead of $Q^+$ for action selection. We will assume the agent will act greedily with respect to its $Q$-value estimates and break ties uniformly:

$$a_t \leftarrow \text{Uniform}\{\arg\max_a Q_t(s_t, a)\}.$$

Consider the single state MDP in Figure 17 with $H = 1$. We use this MDP to show that with 0.5 probability pessimistically initialised greedy $Q$-learning never finds the optimal policy.

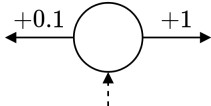

Figure 17: A simple failure case for pessimistically initialised greedy $Q$-learning. There is 1 state with 2 actions and $H = 1$. The agent receives 0.1 reward for the left action and 1 for the right action.

The agent receives a reward of $+1$ for selecting the right action and $0.1$ otherwise. Therefore the optimal policy is to select the right action. Now consider the first episode: $\forall a, Q_1(s, a) = 0$. Thus, the agent selects an action at random with uniform probability. If it selects the left action, it updates:

$$Q_1(s, L) = \eta_1( \underbrace{0.1}_{\text{MDP reward}} + \underbrace{r_{\text{int}}}_{\text{Intrinsic Reward}} + \underbrace{b}_{\text{Bootstrap}} ) > 0.$$

Thus, in the second episode it selects the left action again, since $Q_1(s, L) > 0 = Q_1(s, R)$. Our estimate of $Q_1(s, L)$ never drops below $0.1$, and so the right action is never taken. Thus, with probability of $\frac{1}{2}$ it never selects the correct action (also a linear regret of $0.9T$).

This counterexample applies for any non-negative intrinsic motivation (including no intrinsic motivation), and is unaffected if we utilise optimistic bootstrapping or not.

## H NECESSITY FOR INTRINSIC MOTIVATION BONUS

Despite introducing an extra optimism term with a tunable hyperparameter $M$, OPIQ still requires the intrinsic motivation term $b_i^T$ to ensure it does not under-explore in stochastic environments.

We will prove that OPIQ without the intrinsic motivation term $b_i^T$ does not satisfy Theorem 1. Specifically we will show that there exists a 1 state, 2 action MDP with stochastic reward function such that for all $M > 0$ the probability of incurring linear regret is greater than the allowed failure probability $p$. We choose to use stochastic rewards as opposed to stochastic transitions for a simpler proof.

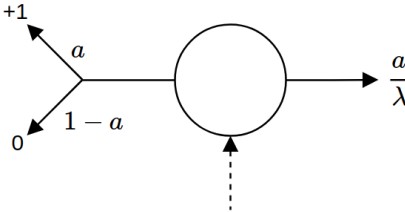

Figure 18: The parametrised MDP.

The MDP we will use is shown in Figure 18, where $\lambda > 1$ and $a \in (0,1)$ s.t $p < 1 - a$. $H = 1, S = 1$ and $A = 2$. The reward function for the left action is stochastic, and will return $+1$ reward with probability $a$ and $0$ otherwise. The reward for the right action is always $a/\lambda$.

Let $p > 0$, the probability with which we are allowed to incur a total regret not bounded by the theorem. OPIQ cannot depend on the value of $\lambda$ or $a$ as they are unknown.

Pick $\lambda$ s.t $M > \frac{\log(\frac{\lambda}{a})}{\log(\frac{\log(p)}{\log(1-a)})}$.

OPIQ will recover the sub-optimal policy of taking the right action if every time we take the left action we receive a $0$ reward. This will happen since our $Q^+$-value estimate for the left action will eventually drop below the $Q$-value estimate for the right action which is $a/\lambda > 0$. The sup-optimal policy will incur a linear regret, which is not bounded by the theorem.

Our probability of failure is at least $(1-a)^R$, where $R$ is the number of times we select the left action, which decreases as $R$ increases. This corresponds to receiving a $0$ reward for every one of the $R$ left transitions we take. Note that $(1-a)^R$ is an underestimate of the probability of failure.

For the first 2 episodes we will select both actions, and with probability $(1-a)$ the left action will return $0$ reward. Our $Q$-values will then be: $Q_1(s,L) = 0, Q_1(s,R) = a/\lambda$.

It is possible to take the left action as long as

$$\frac{1}{(R+1)^M} \geq \frac{a}{\lambda},$$

since the optimistic bonus for the right action decays to $0$.

This then provides a very loose upper bound for $R$ as $(\frac{\lambda}{a})^{1/M}$, which then leads to a further underestimation of the probability of failure.

Assume for a contradiction that $(1-a)^R < p$:

$$
\begin{aligned}
(1-a)^R < p &\iff (1-a)^{(\lambda/a)^{1/M}} < p \\
&\iff (\lambda/a)^{1/M} \log(1-a) < \log(p) \\
&\iff (\lambda/a)^{1/M} > \log(p)/\log(1-a) \\
&\iff (1/M)\log(\lambda/a) > \log(\log(p)/\log(1-a)) \\
&\iff M < \log(\lambda/a)/\log(\log(p)/\log(1-a))
\end{aligned}
$$

(6)

This provides our contradiction as we choose $\lambda$ s.t $M > \log(\lambda/a)/\log(\log(p)/\log(1-a))$. We can always pick such a $\lambda$ because $\log(\lambda/a)$ can get arbitrarily close to $0$.

So our probability of failure (of which $(1-a)^R$ is a severe underestimate) is greater than the allowed probability of failure $p$.

# I  PROOF OF THEOREM 1

**Theorem 1.** *For any $p \in (0,1)$, with probability at least $1 - p$ the total regret of $Q^+$ is at most $\mathcal{O}(\sqrt{H^4 SAT \log(SAT/p)})$ for $M \geq 1$ and at most $\mathcal{O}(H^{1+M} SAT^{1-M} + \sqrt{H^4 SAT \log(SAT/p)})$ for $0 < M < 1$.*

OPIQ is heavily based on UCB-H (Jin et al., 2018), and as such the proof very closely mirrors its proof except for a few minor differences. For completeness, we reproduce the entirety of the proof with the minor adjustments required for our scheme.

The proof is concerned with bounding the regret of the algorithm after $K$ episodes. The algorithm we follow takes as input the value of $K$, and changes the magnitudes of $b_N^T$ based on it.

We will make use of a corollary to Azuma's inequality multiple times during the proof.

**Theorem 2.** *(Azuma's Inequality). Let $Z_0, ..., Z_n$ be a martingale sequence of random variables such that $\forall i \exists c_i : |Z_i - Z_{i-1}| < c_i$ almost surely, then:*

$$P(Z_n - Z_0 \geq t) \leq \exp \frac{-t^2}{2 \sum_{i=1}^n c_i^2}$$

**Corollary 1.** *Let $Z_0, ..., Z_n$ be a martingale sequence of random variables such that $\forall i \exists c_i : |Z_i - Z_{i-1}| < c_i$ almost surely, then with probability at least $1 - \delta$:*

$$|Z_n - Z_0| \leq \sqrt{2(\sum_{i=1}^n c_i^2) \log \frac{2}{\delta}}$$

**Lemma 1.** *(Jin et al., 2018)*
*Define $\eta_N^0 = \prod_{j=1}^N (1 - \eta_j), \eta_N^i = \eta_i \prod_{j=i+1}^N (1 - \eta_j)$*
*The following properties hold for $\eta_N^i$:*

- $\frac{1}{\sqrt{N}} \leq \sum_{i=1}^N \frac{\eta_N^i}{\sqrt{i}} \leq \frac{2}{\sqrt{N}}, \forall N \geq 1$

- $\max_{i=1,...,N} \eta_N^i \leq \frac{2H}{N}, \sum_{i=1}^N (\eta_N^i)^2 \leq \frac{2H}{N}, \forall N \geq 1$

- $\sum_{N=i}^\infty \eta_N^i = 1 + \frac{1}{H}, \forall i \geq 1$

**Lemma 2.** *Adapted slightly from (Jin et al., 2018)*
*Define $V_t^k(s) := \min\{H, \max_{a'} Q_t^{+,k}(s,a')\}, \forall s \in S$.*
*For notational convenience, we also define $[P_t^k V_{t+1}](s_t^k, a_t^k) := V_{t+1}(s_{t+1}^k)$, where $s_t^k$ was the state encountered at episode $k$ at timestep $t$ (similarly for $a_t^k$),*
*and $[P V_{t+1}](s,a) := \mathbb{E}_{s' \sim P_t(\cdot | s_t = s, a_t = a)} [V_{t+1}(s')]$.*

*For any $(s,a,t) \in \mathcal{S} \times \mathcal{A} \times [H]$, episode $k \leq K$ and $N = N(s,a,t)$. Suppose $(s,a)$ was previously taken at step $t$ of episodes $k_1, ..., k_N < k$. Then:*

$$(Q_t^{+,k} - Q_t^*)(s,a) = -\eta_N^0 Q_t^*(s,a)$$
$$+ \sum_{i=1}^N \eta_N^i [(V_{t+1}^{k_i} - V_{t+1}^*)(s_{t+1}^{k_i}) + (P_t - P_t^{k_i}) V_{t+1}^*(s,a) + b_i^T] + \frac{H}{(N+1)^M}$$

*Proof.* We have the following recursive formula for $Q^+$ at episode $k$ and timestep $t$:

$$Q_t^{+,k}(s,a) = \sum_{i=1}^N \eta_N^i [r_t(s,a) + V_{t+1}^{k_i}(s_{t+1}^{k_i}) + b_i^T] + \frac{H}{(N+1)^M} \tag{7}$$

We can produce a similar formula for $Q^*$:

$$Q_t^*(s,a) = (r_t + P_t(s,a) V_{t+1}^*)(s,a)$$

From the Bellman Optimality Equation

$$= \eta_N^0 Q_t^*(s,a) + \sum_{i=1}^{N} \eta_N^i [r_t(s,a) + P_t V_{t+1}^*(s,a)]$$

Since $\sum_{i=1}^{N} \eta_N^i = 1$ and $\eta_N^0 = 0$ for $N \geq 1$ and $\sum_{i=1}^{N} \eta_N^i = 0$ and $\eta_N^0 = 1$ for $N = 0$

$$= \eta_N^0 Q_t^*(s,a) + \sum_{i=1}^{N} \eta_N^i [r_t(s,a) + (P_t - P_t^{k_i})V_{t+1}^*(s,a) + P_t^{k_i} V_{t+1}^*(s,a)]$$

$\pm P_t^{k_i} V_{t+1}^*(s,a)$ inside the summation.

$$= \eta_N^0 Q_t^*(s,a) + \sum_{i=1}^{N} \eta_N^i [r_t(s,a) + (P_t - P_t^{k_i})V_{t+1}^*(s,a) + V_{t+1}^*(s_{t+1}^{k_i})] \qquad (8)$$

By definition of $P_t^k$

Subtracting equation 7 from equation 8 gives the required result.

$\square$

**Lemma 3.** *Adapted slightly from (Jin et al., 2018)*
*Bounding $Q^+ - Q^*$*

*There exists an absolute constant $c > 0$ such that, for any $\delta \in (0,1)$, letting $b_N^T = 2\sqrt{\frac{H^3 \log(SAT/\epsilon)}{N}}$,*
*we have that $\beta_N^T = 2\sum_{i=1}^{N} \eta_N^i b_i^T \leq 8\sqrt{\frac{H^3 \log(SAT/\epsilon)}{N}}$ and, with probability at least $1 - \delta$, the*
*following holds simultaneously for all $(s,a,t,k) \in \mathcal{S} \times \mathcal{A} \times [H] \times [K]$:*

$$0 \leq (Q_t^{+,k} - Q_t^*)(s,a) \leq \sum_{i=1}^{N} \eta_N^i [(V_{t+1}^{k_i} - V_{t+1}^*)(s_{t+1}^{k_i})] + \beta_N^T + \frac{H}{(N+1)^M}$$

*where $N = N(s,a,t)$ and $k_1, ..., k_N < k$ are the episodes where $(s,a)$ was taken at step $t$.*

*Proof.* For each fixed $(s,a,t) \in \mathcal{S} \times \mathcal{A} \times [H]$, let $k_0 = 0$ and

$$k_i = \min(\{k \in [K] | k > k_{i-1}, (s_t^k, a_t^k) = (s,a)\} \cup \{K+1\})$$

$k_i$ is then the episode at which $(s,a)$ was taken at step $t$ for the $i$th time, or $k_i = K+1$ if it has been taken fewer than $i$ times.

Then the random variable $k_i$ is a *stopping time*. Let $\mathcal{F}_i$ be the $\sigma$-field generated by all the random variables until episode $k_i$ step $t$.

Let $\tau \in [K]$.

Let $X_i := \eta_\tau^i \mathbb{1}[k_i \leq K] [(P_t^{k_i} - P_t)V_{t+1}^*](s,a))$. Then $Z_i = \sum_{j=1}^{i} X_i$ is also a martingale sequence with respect to the filtration $(\mathbb{1}[k_i \leq K] [(P_t^{k_i} - P_t)V_{t+1}^*](s,a))_{i=1}^{\tau}$, $Z_0 = 0$, $Z_n - Z_0 = \sum_{i=1}^{n} X_i$ and $Z_i - Z_{i-1} = X_i$. We also have that $|X_i| \leq \eta_\tau^i H$.

Then by Azuma's Inequality we have that with probability at least $1 - 2\delta/(SAHK)$

$$|\sum_{i=1}^{\tau} \eta_\tau^i \mathbb{1}[k_i \leq K] [(P_t^{k_i} - P_t)V_{t+1}^*](s,a)| \leq \sqrt{2(\sum_{i=1}^{\tau}(\eta_\tau^i H)^2) \log \frac{SAT}{\delta}}$$

$$= H\sqrt{2(\sum_{i=1}^{\tau}(\eta_\tau^i)^2) \log \frac{SAT}{\delta}}$$

Then by a Union bound over all $\tau \in [K]$, we have that with probability at least $1 - 2\delta/(SAH)$:

$$\forall \tau \in [K] \mid \sum_{i=1}^{\tau} \eta_\tau^i \mathbb{1}[k_i \leq K] \left[(P_t^{k_i} - P_t)V_{t+1}^*\right](s,a)| \leq H\sqrt{2(\sum_{i=1}^{\tau}(\eta_\tau^i)^2) \log \frac{SAT}{\delta}}$$

$$\leq 2\sqrt{\frac{H^3 \log SAT/\delta}{\tau}} \qquad (9)$$

Since From Lemma 1 we have that $\sum_{i=1}^{\tau}(\eta_N^i)^2 \leq \frac{2H}{\tau}$

Since inequality equation 9 holds for all fixed $\tau \in [K]$ uniformly, it also holds for a random variable $\tau$ $\tau = N = N^k(s,a,t) \leq K$. Also note that $\mathbb{1}[k_i \leq K] = 1$ for all $i \leq N$.

We can then additionally apply a union bound over all $s \in \mathcal{S}, a \in \mathcal{A}, t \in [H]$ to give:

$$|\sum_{i=1}^{N} \eta_N^i[(P_t^{k_i} - P_t)V_{t+1}^*](s,a)| \leq 2\sqrt{\frac{H^3 \log SAT/\delta}{N}} = b_N^T \qquad (10)$$

which holds with probability $1 - 2\delta$ for all $(s,a,t,k) \in \mathcal{S} \times \mathcal{A} \times [H] \times [K]$. We then rescale $\delta$ to $\delta/2$.

By Lemma 1 we have that $b_N^T = 2\sqrt{H^3 \log(SAT/\delta)/N} \leq \beta_N^T/2 = \sum_{i=1}^{N} \eta_N^i b_i^T \leq 4\sqrt{H^3 \log(SAT/\delta)/N} = 2b_N^T$.

From Lemma 2 we have that:

$$(Q_t^{+,k} - Q_t^*)(s,a) = -\eta_N^0 Q_t^*(s,a)$$

$$+ \sum_{i=1}^{N} \eta_N^i[(V_{t+1}^{k_i} - V_{t+1}^*)(s_{t+1}^{k_i}) + (P_t - P_t^{k_i})V_{t+1}^*(s,a) + b_i^T] + \frac{H}{(N+1)^M}$$

$$= -\eta_N^0 Q_t^*(s,a) + \sum_{i=1}^{N} \eta_N^i[(V_{t+1}^{k_i} - V_{t+1}^*)(s_{t+1}^{k_i})]$$

$$+ \sum_{i=1}^{N} \eta_N^i[(P_t - P_t^{k_i})V_{t+1}^*(s,a)] + \sum_{i=1}^{N} \eta_N^i[b_i^T] + \frac{H}{(N+1)^M}$$

rearranging terms

$$\leq \sum_{i=1}^{N} \eta_N^i[(V_{t+1}^{k_i} - V_{t+1}^*)(s_{t+1}^{k_i})] + b_N^T + \beta_N^T/2 + \frac{H}{(N+1)^M}$$

From Eqn equation 10, defn on $\beta_N^T$ and non-negativity of $Q^*$

$$\leq \sum_{i=1}^{N} \eta_N^i[(V_{t+1}^{k_i} - V_{t+1}^*)(s_{t+1}^{k_i})] + \beta_N^T + \frac{H}{(N+1)^M}$$

Since $b_N^T \leq \beta_N^T/2$

which gives the R.H.S.

Eqn equation 10 tells us that $\sum_{i=1}^{N} \eta_N^i[(P_t - P_t^{k_i})V_{t+1}^*](s,a) \geq -b_N^T$, along with $b_N^T \leq \beta_N^T/2 = \sum_{i=1}^{N} \eta_N^i b_i^T$ which then gives:

$$(Q_t^{+,k} - Q_t^*)(s,a) = -\eta_N^0 Q_t^*(s,a)$$

$$+ \sum_{i=1}^{N} \eta_N^i[(V_{t+1}^{k_i} - V_{t+1}^*)(s_{t+1}^{k_i}) + (P_t - P_t^{k_i})V_{t+1}^*(s,a) + b_i^T] + \frac{H}{(N+1)^M}$$

$$\geq -\eta_N^0 Q_t^*(s,a) + \sum_{i=1}^{N} \eta_N^i[(V_{t+1}^{k_i} - V_{t+1}^*)(s_{t+1}^{k_i})] + \frac{H}{(N+1)^M}$$

We can then prove the L.H.S. by induction on $t = H, ..., 1$. For $t = H(Q_t^{+,k} - Q_t^*) \geq 0$. This is because $V_{H+1}^{k_i} = V_{H+1}^* = 0$, for $N = 0 \frac{H}{(0+1)^M} = H > Q_H^*$, and for $N > 0$ we have that $\eta_N^0 = 0$. If we assume the statement true for $t + 1$, consider $t$:

$$(Q_t^{+,k} - Q_t^*)(s,a) \geq -\eta_N^0 Q_t^*(s,a) + \sum_{i=1}^N \eta_N^i [(V_{t+1}^{k_i} - V_{t+1}^*)(s_{t+1}^{k_i})] + \frac{H}{(N+1)^M}$$

$$= -\eta_N^0 Q_t^*(s,a) + \sum_{i=1}^N \eta_N^i [\min\{H, \max_{a'} Q_{t+1}^{+,k_i}(s_{t+1}^{k_i}, a')\} - \max_{a''} Q_{t+1}^*(s_{t+1}^{k_i}, a'')]$$

$$+ \frac{H}{(N+1)^M}$$

$$\geq -\eta_N^0 Q_t^*(s,a) + \frac{H}{(N+1)^M}$$

If $\min\{H, \max_{a'} Q_{t+1}^{+,k_i}(s_{t+1}^{k_i}, a')\} = \max_{a'} Q_{t+1}^{+,k_i}(s_{t+1}^{k_i}, a')$, by our inductive assumption since $(Q_{t+1}^{+,k} - Q_{t+1}^*)(s,a) \geq 0 \implies \max_{a'} Q_{t+1}^{+,k_i}(s, a') \geq \max_{a''} Q_{t+1}^*(s, a'')$

If $\min\{H, \max_{a'} Q_{t+1}^{+,k_i}(s_{t+1}^{k_i}, a')\} = H$, we have that $\max_{a''} Q_{t+1}^*(s_{t+1}^{k_i}, a'') \leq H$.

This then proves the L.H.S.

$\square$

***Note on stochastic rewards:*** *We have assumed so far that the reward function is deterministic for a simpler presentation of the proof. If we allowed for a stochastic reward function, the previous lemmas can be easily adapted to allow for it.*

*Lemma 2 would give us:*

$$(Q_t^{+,k} - Q_t^*)(s,a) = -\eta_N^0 Q_t^*(s,a) + \sum_{i=1}^N \eta_N^i [(r_t^{k_i} - \mathbb{E}_{r' \sim R_t(\cdot|s,a)}[r'])$$

$$+ (V_{t+1}^{k_i} - V_{t+1}^*)(s_{t+1}^{k_i}) + (P_t - P_t^{k_i})V_{t+1}^*(s,a) + b_i^T] + \frac{H}{(N+1)^M}$$

$(r_t^{k_i} - \mathbb{E}_{r' \sim R_t(\cdot|s,a)}[r'])$ *can then be bounded in the same way as in Lemma 3. Increasing the constants for $b_N^T, \beta_N^T$ and rescaling of $\delta$ appropriately then completes the necessary changes.*

*We will now prove that our algorithm is provably efficient. We will do this by showing it has sub-linear regret, following closely the proof of Theorem 1 from (Jin et al., 2018).*

**Theorem 1.** *For any $p \in (0,1)$, with probability at least $1 - p$ the total regret of $Q^+$ is at most $\mathcal{O}(\sqrt{H^4 SAT \log(SAT/p)})$ for $M \geq 1$ and at most $\mathcal{O}(H^{1+M}SAT^{1-M} + \sqrt{H^4 SAT \log(SAT/p)})$ for $0 < M < 1$.*

*Proof.* Let

$$\delta_t^k := (V_t^k - V_t^{\pi_k})(s_t^k), \phi_t^k := (V_t^k - V_t^*)(s_t^k)$$

Lemma 3 tells us with probability at least $1 - \delta$ that $Q_t^{+,k} \geq Q_t^*$, which also implies that $V_t^k \geq V_t^*$. We can then upperbound the regret as follows:

$$\text{Regret}(K) = \sum_{k=1}^K (V_1^* - V_1^{\pi_k})(s_1^k) \leq \sum_{k=1}^K (V_1^k - V_1^{\pi_k})(s_1^k) = \sum_{k=1}^K \delta_1^k$$

We then aim to bound $\sum_{k=1}^K \delta_t^k$ by the next timestep $\sum_{k=1}^K \delta_{t+1}^k$, which will give us a recursive formula to upperbound the regret. We will accomplish this by relating $\sum_{k=1}^K \delta_t^k$ to $\sum_{k=1}^K \phi_t^k$.

For any fixed $(k, t) \in [K] \times [H]$, let $N_k = N(s_t^k, a_t^k, t)$ where $(s_t^k, a_t^k)$ was previously taken at step $t$ of episodes $k_1, ..., k_N < k$. We then have:

$$\delta_t^k = (V_t^k - V_t^{\pi_k})(s_t^k) \leq (Q_t^{+,k} - Q_t^{\pi_k})(s_t^k, a_t^k)$$
$$= (Q_t^{+,k} - Q_t^*)(s_t^k, a_t^k) + (Q_t^* - Q_t^{\pi_k})(s_t^k, a_t^k)$$
$$\leq \sum_{i=1}^{N_k} \eta_{N_k}^i [(V_{t+1}^{k_i} - V_{t+1}^*)(s_{t+1}^{k_i})]$$
$$+ \beta_{N_k}^T + \frac{H}{(N_k + 1)^M} + [P_t(V_{t+1}^* - V_{t+1}^{\pi_k})](s_t^k, a_t^k)$$

By Lemma 3 for the first term. Bounding $(Q_t^* - Q_t^{\pi_k})$ is acheived through the Bellman Equation giving the final term.

$$= \sum_{i=1}^{N_k} \eta_{N_k}^i \phi_{t+1}^{k_i} + \beta_{N_k}^T + \frac{H}{(N_k + 1)^M} + [P_t(V_{t+1}^* - V_{t+1}^{\pi_k})](s_t^k, a_t^k)$$
$$+ [P_t^k(V_{t+1}^* - V_{t+1}^{\pi_k})](s_t^k, a_t^k) - [P_t^k(V_{t+1}^* - V_{t+1}^{\pi_k})](s_t^k, a_t^k)$$

$\pm [P_t^k(V_{t+1}^* - V_{t+1}^{\pi_k})](s_t^k, a_t^k)$ and subbing for $\phi$

$$= \sum_{i=1}^{N_k} \eta_{N_k}^i \phi_{t+1}^{k_i} + \beta_{N_k}^T + \frac{H}{(N_k + 1)^M} + [(P_t - P_t^k)(V_{t+1}^* - V_{t+1}^{\pi_k})](s_t^k, a_t^k)$$
$$+ (V_{t+1}^* - V_{t+1}^{\pi_k})(s_{t+1}^k)$$

Since $[P_t^k(V_{t+1}^* - V_{t+1}^{\pi_k})](s_t^k, a_t^k) = (V_{t+1}^* - V_{t+1}^{\pi_k})(s_{t+1}^k)$

$$= \sum_{i=1}^{N_k} \eta_{N_k}^i \phi_{t+1}^{k_i} + \beta_{N_k}^T + \frac{H}{(N_k + 1)^M} + \xi_{t+1}^k + \delta_{t+1}^k - \phi_{t+1}^k \qquad (11)$$

Letting $\xi_{t+1}^k := [(P_t - P_t^k)(V_{t+1}^* - V_{t+1}^{\pi_k})](s_t^k, a_t^k).(V_{t+1}^* - V_{t+1}^{\pi_k})(s_{t+1}^k) = \delta_{t+1}^k - \phi_{t+1}^k$.

We must now bound the summation $\sum_{k=1}^K \delta_t^k$, which we will do by considering each term of equation 11 separately.

$\sum_{k=1}^K \frac{H}{(N_k+1)^M}$:

$$\sum_{k=1}^K \frac{H}{(N_k + 1)^M} \geq \sum_{k=1}^K \frac{H}{(K + 1)^M} = \frac{HK}{(K + 1)^M} \implies \sum_{k=1}^K \frac{H}{(N_k + 1)^M} \in \Omega(HK^{1-M})$$

The first inequality follows since $N \leq K$. This shows that we require $M > 0$ in order to guarantee a sublinear regret in $K$.

$$\sum_{k=1}^K \frac{H}{(N_k + 1)^M} = \sum_{k=1}^K \sum_{n=0}^\infty \mathbb{1}\{N_k = n\} \frac{H}{(n + 1)^M} = \sum_{n=0}^\infty \sum_{k=1}^K \mathbb{1}\{N_k = n\} \frac{H}{(n + 1)^M}$$
$$\leq SA \sum_{n=0}^\infty \frac{H}{(n + 1)^M} = SAH \sum_{n=1}^K \frac{1}{n^M}$$

The first equality follows by rewriting $\frac{H}{(N_k+1)^M}$ as $\sum_{n=0}^\infty \mathbb{1}\{N_k = n\} \frac{H}{(n+1)^M}$. Only one of the indicator functions will be true for episode $k$, giving us the required value. The inequality is a crude upper bound that suffices for the proof. For a given $n$, $\mathbb{1}\{N_k = n\}$ can only be true for at most $SA$ times across all of training. They will contribute $\frac{H}{(n+1)^M}$ to the final sum.

For $M > 1$, the sum $\sum_{n=1}^K \frac{1}{n^M}$ is bounded, which means that $\sum_{k=1}^K \frac{H}{(N_k+1)^M} \in \mathcal{O}(SAH)$

For $M = 1$, $\sum_{n=1}^{K} \frac{1}{n^M} \le \log(K+1) \implies \sum_{k=1}^{K} \frac{H}{(N_k+1)^M} \in \mathcal{O}(SAH\log(K))$

For $0 < M < 1$, $\sum_{k=1}^{K} \frac{H}{(N_k+1)^M} \in \mathcal{O}(SAHK^{1-M})$, since $\sum_{n=1}^{K} \frac{1}{n^M} \sim \frac{K^{1-M}}{1-M}$ (Goel and Rodriguez, 1987)

$\sum_{k=1}^{K} \sum_{i=1}^{N_k} \eta_{N_k}^i \phi_{t+1}^{k_i}$:

$$\sum_{k=1}^{K} \sum_{i=1}^{N_k} \eta_{N_k}^i \phi_{t+1}^{k_i} \le \sum_{k'=1}^{K} \phi_{t+1}^{k'} \sum_{i=N_{k'}+1}^{\infty} \eta_i^{N_{k'}} \le (1 + \frac{1}{H}) \sum_{k=1}^{K} \phi_{t+1}^k$$

The first inequality is achieved by rearranging how we take the sum.

Consider a $k' \in [K]$. The term $\phi_{t+1}^{k'}$ will appear in the summation for $k > k'$ provided that $(s_t^k, a_t^k) = (s_t^{k'}, a_t^{k'})$, e.g. we took the same state-action pair at episodes $k$ and $k'$.

On the first occasion, the associated learning rate will be $\eta_{N_{k'}+1}^{N_{k'}}$, on the second $\eta_{N_{k'}+2}^{N_{k'}}$.

We can then consider all possible counts up to $\infty$ to achieve an inequality. By considering all $k' \in [K]$ we will not miss any terms of the original summation.

The second inequality follows by application of Lemma 1 on the sum involving the learning rate.

$\sum_{k=1}^{K} \beta_{N_k}$:

For all $t \in [H]$, we have that:

$$\sum_{k=1}^{K} \beta_{N_k} \le \sum_{k=1}^{K} 8\sqrt{\frac{H^3 \log(SAT/\delta)}{N_k}} = \sum_{s,a} \sum_{n=1}^{N_k(s,a,t)} 2\sqrt{\frac{H^3 \log(SAT/\delta)}{n}}$$

$$\le \sum_{s,a} \sum_{n=1}^{N_k(s,a,t)=\frac{K}{SA}} 2\sqrt{\frac{H^3 \log(SAT/\delta)}{n}}$$

$$\le \sum_{s,a} 2\sqrt{H^3 \log(SAT/\delta)} \sum_{n=1}^{\frac{K}{SA}} \frac{1}{\sqrt{n}}$$

$$\le 4SA\sqrt{H^3 \log(SAT/\delta)}\sqrt{\frac{K}{SA}}$$

$$\in \mathcal{O}(\sqrt{H^3 SAK \log(SAT/\delta)})$$

$$= \mathcal{O}(\sqrt{H^2 SAT \log(SAT/\delta)})$$

We transform the sum over the episode's counts $N_k$ into a sum over all state-action pairs, summing over all of their respective counts.

The first inequality is due to $N_k = K/SA$ maximising the quantity, this is since $1/\sqrt{n}$ grows smaller as $n$ increases. Thus, we want to *spread out* the summation across the state-action space as much as possible.

Then we use the result that $\sum_{i=1}^{n} 1/\sqrt{i} \le 2\sqrt{n}$, and that we are taking the sum over $SA$ identical terms. We then set $n = K/SA$ and are thus taking the sum over $N_k(s, a, t)$ identical terms.

$\sum_{t=1}^{H} \sum_{k=1}^{K} \xi_{t+1}^k$

Following a similar argument as from Lemma 3, we can apply Azuma's inequality and a union bound over all $(s, a, t) \in S \times A \times [H]$. Let $X_i = [(P_t - P_t^k)(V_{t+1}^* - V_{t+1}^{\pi_k})](s_t^k, a_t^k)$. Then $Z_i := \sum_{j=1}^{i} X_j$ is a martingale difference sequence, and we have that $|Z_i - Z_{i-1}| = |X_i| \le H$, and $Z_0 = 0$. Then with probability at least $1 - 2\delta$ we have that:

$$|\sum_{t=1}^{H} \sum_{k=1}^{K} \xi_{t+1}^k| \le H|\sum_{k=1}^{K} \xi_{t+1}^k| \le H\sqrt{2KH^2 \log(SAT/\delta)} \in \mathcal{O}(H^{3/2}\sqrt{T \log(SAT/\delta)})$$

Finally, we can utilise these intermediate results when bounding the regret via Equation equation 11:

$$\sum_{k=1}^{K} \delta_t^k \leq \sum_{i=1}^{N_k} \eta_{N_k}^i \phi_{t+1}^{k_i} + \beta_{N_k}^T + \frac{H}{(N_k+1)^M} + \xi_{t+1}^k + \delta_{t+1}^k - \phi_{t+1}^k$$

$$\leq (1 + \frac{1}{H}) \sum_{k=1}^{K} \phi_{t+1}^k - \sum_{k=1}^{K} \phi_{t+1}^k + \sum_{k=1}^{K} \delta_{t+1}^k + \sum_{k=1}^{K} (\beta_{N_k}^T + \xi_{t+1}^k) + \sum_{k=1}^{K} \frac{H}{(N_k+1)^M}$$

From our intermediate result on $\phi$

$$\leq (1 + \frac{1}{H}) \sum_{k=1}^{K} \delta_{t+1}^k + \sum_{k=1}^{K} (\beta_{N_k}^T + \xi_{t+1}^k + \frac{H}{(N_k+1)^M})$$

Since $V^* \geq V^{\pi_k} \implies \delta_{t+1}^k \geq \phi_{t+1}^k$

Letting $A_t := \sum_{k=1}^{K} (\beta_{N_k}^T + \xi_{t+1}^k + \frac{H}{(N_k+1)^M})$, we have that $\sum_{k=1}^{K} \delta_H^k \leq A_H$ since $\delta_{H+1} = 0$. By induction on $t = H, ..., 1$ we then have that $\sum_{k=1}^{K} \delta_t^k \leq \sum_{j=0}^{H-t} (1 + \frac{1}{H})^j A_{H-j} \implies \sum_{k=1}^{K} \delta_1^k \in \mathcal{O}(\sum_{t=1}^{H} \sum_{k=1}^{K} (\beta_{N_k}^T + \xi_{t+1}^k + \frac{H}{(N_k+1)^M}))$ since $(1 + 1/H)^H \leq e$.

Using our intermediate results for the relevant quantities then gives us with probability at least $1 - 2\delta$:

$$\sum_{k=1}^{K} \delta_1^k \in \mathcal{O}(\sqrt{(H^4 SAT \log(SAT/\delta))} + H^{3/2} \sqrt{T \log(SAT/\delta)}) + \mathcal{O}(\sum_{t=1}^{H} \sum_{k=1}^{K} \frac{H}{(N_k+1)^M})$$

$$= \mathcal{O}(\sqrt{(H^4 SAT \log(SAT/\delta))}) + \mathcal{O}(\sum_{t=1}^{H} \sum_{k=1}^{K} \frac{H}{(N_k+1)^M})$$

We then consider the 3 cases of $M$:

For $M > 1$ we get $\mathcal{O}(\sqrt{(H^4 SAT \log(SAT/\delta))} + H^2 SA)$.

For $M = 1$ we get $\mathcal{O}(\sqrt{(H^4 SAT \log(SAT/\delta))} + H^2 SA \log(K))$.

And for $0 < M < 1$ we have $\mathcal{O}(\sqrt{(H^4 SAT \log(SAT/\delta))} + H^{1+M} SAT^{1-M})$.

$T \geq \sqrt{H^4 SAT \log(SAT/\delta)} \implies \sqrt{H^4 SAT \log(SAT/\delta)} \geq H^2 SA \log(K)$, which means we can remove the $H^2 SA \log(K)$ or $H^2 SA$ from the upperbound. If $T \leq \sqrt{H^4 SAT \log(SAT/\delta)}$ then that is also a sufficient upper bound since the regret cannot be more than $HK = T$.

This gives us the required result after rescaling $\delta$ to $\delta/2$.

$\square$

