# OpenReview forum: "Optimistic Exploration even with a Pessimistic Initialisation"
_ICLR.cc/2020/Conference — Accept (Poster)_

### Official Review · AnonReviewer2 · 2019-10-17
**Official Blind Review #2**

**Rating:** 6

**Review:**

Optimistic Exploration even with a Pessimistic Initialisation
================================================================

This paper presents an exploration algorithm based on "optimism in the face of uncertainty" via count-based bonus.
The authors observe that typical neural net initializations close to zero can be pessimistic, but show that augmenting a count-based bonus for acting and bootstrapping can overcome this.
The authors support their claim with an adaptation of a regret bound for the tabular case, and a series of didactic experiments with neural net models.


There are several things to like about this paper:
- Exploration with generalization in Deep RL is a large outstanding problem, with few effective options and none really commonly used in the field beyond epsilon-greedy or Boltzmann.
- This algorithm is reasonably well thought out, building on an established literature of exploration bonuses, but with a slightly different take on the structure of the bonus.
- The paper is well structured, building from intuition to theory to toy examples in DQN setting.
- Overall the algorithm appears to perform well against a wide variety of related variants (although that presentation is a little confusing / overwhelming).


There are several places the paper might be improved:
- I don't think the authors make a clear enough case for why this method is *better* than the other optimistic bonus approaches listed... Yes there is a regret bound, but this is not as good as some other methods... Yes there are ablations... but they're not really clear about what the mechanism that makes this method better than others!
- Although this algorithm is motivated by applications to *deep* RL, the key choice of the "count" (and thus the method for optimism bonus) is mostly sidestepped. It amounts to an essentially tabular bonus in the space of the hashing function... and it's not clear why this approach should work any better or worse than other similar approaches that the paper complains about. For example, if you used "Randomized Prior Functions" or "Random Network Distillation" with that same hashing functions you would likely end up with similar results?
- The comparison to benchmark algorithms seems quite confusing and I'm not sure if it's really presented well. It might be good to focus on fewer comparisons at a time and push remaining less important ones to the appendix.
- It would be great to get an evaluation of these algorithms on a standardized and open-source benchmark... and I think that bsuite could be a really good candidate for this paper https://github.com/deepmind/bsuite particularly the "deep sea" experiments.


Overall I think this is a reasonable paper, and I expect it to improve during the review process.
At the moment I have to say that I don't think there is a clear enough case for why this method is preferable to other similar approaches, or enough insight into the pros and cons to accept.

============

Updating to "weak accept" as part of rebuttal.

**Experience Assessment:**

I have published in this field for several years.

**Review Assessment: Checking Correctness Of Derivations And Theory:**

I assessed the sensibility of the derivations and theory.

**Review Assessment: Checking Correctness Of Experiments:**

I carefully checked the experiments.

**Review Assessment: Thoroughness In Paper Reading:**

I read the paper at least twice and used my best judgement in assessing the paper.

---

> ### Author Response · Authors · 2019-11-11
> **Replying to Reviewer #2**
>
> Thank you for your review.
>
> “I don't think the authors make a clear enough case for why this method is *better* than the other optimistic bonus approaches listed…”
> >OPIQ provides an incentive for the agent to try and explore new state-action pairs by using optimistic Q-Value estimates during action selection and bootstrapping. Simply adding an intrinsic reward in novel states, after visiting them, merely provides an incentive to return to those states, but does not provide any incentive to try a new action during action selection. Additionally, an agent only using intrinsic rewards can prematurely stop exploring states that it has visited a number of times already, even if there are still actions it has never tried before in those states. OPIQ ensures the agent is incentivised to do both of these by explicitly ensuring that optimistic Q-Values are used during action selection and bootstrapping.
> We have updated the Motivations subsection (3.1) to include a paragraph about why optimistic estimates lead to better exploration.
>
> The ablation OPIQ w/o OB (light green in Figures 4 and 5) only differs from a DQN with pseudo-count based intrinsic motivation (DQN + PC) during action selection, but performs significantly better showing the importance of optimism (particularly during action selection in this particular environment).
>
> The regret bound is not designed to be as competitive as possible with other tabular algorithms. It is a sanity check that our modifications to UCB-H in the tabular setting maintain the same efficiency guarantees, and provides a firm theoretical foundation for OPIQ.
>
> “the key choice of the "count" (and thus the method for optimism bonus) is mostly sidestepped”
> > We believe that the shortcomings and limitations that OPIQ aims to address are orthogonal to the choice of pseudocount. There has been a large amount of work proposing new methods for obtaining approximate counts (either directly or through density models) to use as intrinsic motivation bonuses that we build on.
>
> “For example, if you used "Randomized Prior Functions" or "Random Network Distillation" with that same hashing functions you would likely end up with similar results?”
> > Could you please clarify how you intend the hashing function to be used with these approaches?
>
> “The comparison to benchmark algorithms seems quite confusing and I'm not sure if it's really presented well. It might be good to focus on fewer comparisons at a time and push remaining less important ones to the appendix.”
> > The experimental results are indeed quite crowded, but we believe it is important to compare against a wide variety of algorithms to provide strong experimental results.
> We have uploaded a revision of the paper with a revised results section that should address your concerns.
>
> “It would be great to get an evaluation of these algorithms on a standardized and open-source benchmark... and I think that bsuite could be a really good candidate for this paper https://github.com/deepmind/bsuite particularly the "deep sea" experiments.”
> > The “deep sea” environment is very similar to the Chain environment we tested our algorithm on (although bsuite uses a ‘length’ N up to 50, whereas we tested N=100).
> Thank you for the suggestion, we will look into bsuite.
> Additionally, we plan on open-sourcing our code so that the environments can be used in future research.
>
> We have also updated the results section to include some experiments on Montezuma’s Revenge showing that OPIQ is able to scale to these complex environments and continue to provide exploration advantages.

---

> > ### Comment · AnonReviewer2 · 2019-11-14
> > **Clarifying some points**
> >
> > Thank you for your response!
> >
> > I think it's clear that the authors have made an effort to update their paper in response to the reviewers.
> > It will take me some time to go through the new paper fully, but it does look better.
> > My feeling is that I should increase my score to "weak accept" to reflect this, but also will need time to look at it properly.
> >
> > Responding to each point:
> >
> > - I'm still not convinced to this as an improved method for optimism, but I think that the paper now makes a clearer case in terms of the ablations.
> >
> > - The choice of "count" as the baseline for exploration is still not examined. Count seems like a proxy for "uncertainty" which is valid in the tabular setting, but not valid with generalization.
> >
> > - Randomized prior p(state) where each p(state) is for example N(0,1) based on the hash(state).
> >
> > - The experiments do look more clear.
> >
> > - bsuite I think would be a natural fit for this paper, and it would be a nice complementary analysis that *should* be quite easy to run. Opensourcing the code is good, but usually is difficult for people to bring into like for like methodology. I agree that the chain is similar to deep sea, which is why I believe your algorithm *should* be able to perform well... plus we'll get some insight into its *scaling* as you increase the size N.

---

> > > ### Author Response · Authors · 2019-11-15
> > > **Replying to Reviewer**
> > >
> > > Thank you for your comments, and for raising your score.
> > >
> > > "The choice of "count" as the baseline for exploration is still not examined. Count seems like a proxy for "uncertainty" which is valid in the tabular setting, but not valid with generalization."
> > > > Using a different method to produce uncertainty estimates, and using those in place of the counts in OPIQ would be a very interesting avenue for future research. Counts were chosen to stay as close to the tabular theory/inspirations as possible, and because they have demonstrated strong empirical performance (especially on Atari [1,2]).
> > >
> > > "Randomized prior p(state) where each p(state) is for example N(0,1) based on the hash(state)."
> > > > Thanks for clarifying, unfortunately, there was not enough time to run these experiments on the maze environment in time for the rebuttal.
> > >
> > > [1] Bellemare, Marc, et al. "Unifying count-based exploration and intrinsic motivation." Advances in Neural Information Processing Systems. 2016.
> > > [2] Ostrovski, Georg, et al. "Count-based exploration with neural density models." Proceedings of the 34th International Conference on Machine Learning-Volume 70. JMLR. org, 2017.

---

### Official Review · AnonReviewer1 · 2019-10-25
**Official Blind Review #1**

**Rating:** 6

**Review:**

The paper proposes a method to optimistically initialize Q-values for unseen state actions for the case of Deep Q-Networks (DQNs) with high dimensional state representations, in order to step closer towards the optimistic initializations in the tabular Q-learning case which have proven guarantees of convergence. The paper shows that simple alternatives such as adding a bias to a DQN do not help as the generalization to novel states usually reduces the optimism of unvisited state-actions. Instead, a separate optimistic Q function is proposed that is an addition of two parts - a Deep Q-network which may be pessimistically initialized (in the worst case) and a term with pseudo-counts of state-actions that together form an optimistic estimate of novel state-actions. For the tabular case, the paper shows a convergence with guarantees similar to UCB-H (Jin et. al., 2018). This optimistic Q-function is used during action selection as well as bootstrapping in the n-step TD update.

I vote for weak accept as this paper does a great job at demonstrating the motivation, simple examples and thorough comparisons for their proposed “OPIQ” model on simple environments such as the Randomized Chain (from Shyam et. al., 2019) and a 2D maze grid. While the experiments are in toy settings, the connection to UCB-H and the novel optimistic Q-function and it’s training formulation make the contributions of this paper significant.

However, my confidence on this rating is low as I have not gone through the theorem in the appendix and I may be wrong in judging the amount of empirical evidence required for the approach.

While the paper does cover a lot of ground with important details and clear motivation, a lot of the desired experiments have been left to future work as mentioned in the paper. This, in addition to the experiments on just toy settings, is not sufficient to conclude that this approach may be applicable to actually high dimensional state spaces where pseudo counts do not work well. Ultimately, the proposed approach relies strongly on good pseudo-count estimates in high dimensional state spaces, which is still an open problem.


**Experience Assessment:**

I have read many papers in this area.

**Review Assessment: Checking Correctness Of Derivations And Theory:**

I assessed the sensibility of the derivations and theory.

**Review Assessment: Checking Correctness Of Experiments:**

I carefully checked the experiments.

**Review Assessment: Thoroughness In Paper Reading:**

I read the paper at least twice and used my best judgement in assessing the paper.

---

> ### Author Response · Authors · 2019-11-11
> **Replying to Reviewer #1**
>
> Thank you for your review.
>
> “Ultimately, the proposed approach relies strongly on good pseudo-count estimates in high dimensional state spaces, which is still an open problem.”
> > Whilst we agree that producing good pseudo-count estimates in high dimensional state spaces is still an open problem, there has been a considerable amount of work attempting to address this with some good success [1,2,3].
> The aim of this paper is to show the importance of optimism during both action-selection and bootstrapping even when using count-based intrinsic rewards. OPIQ builds upon the existing work on producing and using pseudo-counts for exploration, and leads to significantly improved exploration in sparse reward environments.
> Our contributions in this paper are orthogonal to the problem of producing appropriate pseudo-counts.
>
> “...this approach may be applicable to actually high dimensional state spaces where pseudo counts do not work well.”
> > We have uploaded a revision of the paper containing some experiments on Montezuma’s Revenge showing that OPIQ is able to scale to these complex environments and continue to provide exploration advantages, even with a very simplistic pseudo-counting scheme.
>
> [1] Bellemare, Marc, et al. "Unifying count-based exploration and intrinsic motivation." Advances in Neural Information Processing Systems. 2016.
> [2] Ostrovski, Georg, et al. "Count-based exploration with neural density models." Proceedings of the 34th International Conference on Machine Learning-Volume 70. JMLR. org, 2017.
> [3] Tang, Haoran, et al. "# Exploration: A study of count-based exploration for deep reinforcement learning." Advances in neural information processing systems. 2017.

---

### Official Review · AnonReviewer3 · 2019-10-26
**Official Blind Review #3**

**Rating:** 6

**Review:**

#rebuttal responses
 I am pleased by the authors' responses. Thus I change the score to weak accept.

#review
This paper presented OPIQ, a model-free algorithm that does not rely on an optimistic initialization
to ensure efficient exploration. OPIQ augments the Q-values with a new count-based optimism bonus.
OPIO is ensured with good sample efficiency in the tabular setting. Experimental results show that OPIQ drives
better exploration than DQN variants that utilize a pseudo count-based intrinsic motivation in the randomized chain and the maze environment.

The new optimism bonus is interesting and convincing with a good theoretical guarantee. The paper would be more clear if the authors add a motivating example in a tabular environment. That is, why does this extra optimism bonus help to predict optimistic estimates for novel state and action pairs.

I appreciate that the authors compare extensive DQN variants using count-based explorations. But the experimental results are somewhat weak, as there are no comparison results on hard Atari games, such as freeway and Montezuma's revenge.

I am willing to improve the score if the authors show better motivation or results on Atari games.

**Experience Assessment:**

I have read many papers in this area.

**Review Assessment: Checking Correctness Of Derivations And Theory:**

I assessed the sensibility of the derivations and theory.

**Review Assessment: Checking Correctness Of Experiments:**

I carefully checked the experiments.

**Review Assessment: Thoroughness In Paper Reading:**

I read the paper at least twice and used my best judgement in assessing the paper.

---

> ### Author Response · Authors · 2019-11-11
> **Replying to Reviewer #3**
>
> Thank you for your review.
>
> “The paper would be more clear if the authors add a motivating example in a tabular environment. That is, why does this extra optimism bonus help to predict optimistic estimates for novel state and action pairs.”
> > The MDP in Figure 1, which is explored in more detail in Appendix G, provides intuition for why optimism (particularly during action selection) is extremely important even in such a simple MDP.
>
> If a state-action pair has never been visited, then its count will be 0 and hence its optimism bonus will be high. The Q+ values (Q + Optimism Bonus) for unvisited state-action pairs will then also be high. Crucially, we use these Q+ and not just Q during action selection and bootstrapping which ensures the optimism. Being optimistic during action selection in particular encourages the agent to try actions that it has never tried before. Without it, an agent with pessimistically initialised Q-Values could get stuck taking the wrong action indefinitely even on the simple 2-action MDP from Figure 1.
> We have updated section 3.1 to include these motivations/justifications.
>
> In the Deep RL setting, using these optimistic bonuses during action selection and bootstrapping is the crucial difference between OPIQ and other pseudo-count based methods (DQN + PC).
> Figure 6 shows the Q-Values alone are not optimistic for novel state-action pairs, which leads to insufficient exploration as shown by our experimental results in Figures 3 and 4.
>
> “But the experimental results are somewhat weak, as there are no comparison results on hard Atari games, such as freeway and Montezuma's revenge.”
> > We have uploaded a revision of the paper containing some experiments on Montezuma’s Revenge showing that OPIQ is able to scale to these complex environments and continue to provide exploration advantages.

---

> > ### Comment · AnonReviewer3 · 2019-11-13
> > **Replying to Authors**
> >
> > I am pleased that the authors show the comparison results on Montezuma’s Revenge and show better motivation. I would like to see more results on other hard Atari games, such as freeway and Gravitar.

---

> > > ### Author Response · Authors · 2019-11-15
> > > **Replying to Reviewer**
> > >
> > > Thank you for your comments, we are pleased that you like the new results on Montezuma and find the motivation more compelling.
> > > Freeway can be easily solved by simple epsilon greedy [1, 2], so it doesn’t provide much of a challenge in the way of exploration. Unfortunately, there was not enough time to run experiments on Gravitar in time for the rebuttals.
> > >
> > > [1] Bellemare, Marc, et al. "Unifying count-based exploration and intrinsic motivation." Advances in Neural Information Processing Systems. 2016.
> > > [2] Machado, Marlos C., Marc G. Bellemare, and Michael Bowling. "Count-based exploration with the successor representation." arXiv preprint arXiv:1807.11622 (2018)

---

### Decision · Program_Chairs · 2019-12-19

**Decision:**

Accept (Poster)

**Comment:**

The paper propose a scheme to enable optimistic initialization in the deep RL setting, and shows that it's helpful.

The reviewers agreed that the paper is well-motivated and executed, but had some minor reservations (e.g. about the proposal scaling in practice). In an example of a successful rebuttal two of the reviewers raised their scores after the authors clarified the paper and added an experiment on Montezuma's revenge.

The paper proposes a useful, simple and practical idea on the bridge between tabular and deep RL, and I gladly recommend acceptance.